# Methane and carbon dioxide fluxes over a lake: comparison between eddy covariance, floating chambers and boundary layer method

Kukka-Maaria Erkkilä[1], Ivan Mammarella[1], David Bastviken[2], Tobias Biermann[3], Jouni J. Heiskanen[1], Anders Lindroth[4], Olli Peltola[1], Miitta Rantakari[1,5], Timo Vesala[1,6], and Anne Ojala[6,7]

[1]Department of Physics, University of Helsinki, P.O. Box 68, FI-00014 Helsinki, Finland
[2]Department of Thematic Studies — Environmental Change, Linköping University, Linköping, Sweden
[3]Centre for Environmental and Climate Research, Lund University, Sölvegatan 37, 223 62 Lund, Sweden
[4]Department of Physical Geography and Ecosystem Sciences, Lund University, Sölvegatan 12, 223 62 Lund, Sweden
[5]Department of Environmental Sciences, University of Helsinki, P.O. Box 65, FI-00014 Helsinki, Finland
[6]Department of Forest Sciences, University of Helsinki, P.O. Box 27, FI-00014 Helsinki, Finland
[7]Department of Environmental Sciences, University of Helsinki, Niemenkatu 73, FI-15140 Lahti, Finland
*Correspondence to:* Kukka-Maaria Erkkilä (kukka-maaria.erkkila@helsinki.fi)

**Abstract.** Freshwaters bring a notable contribution to the global carbon budget by emitting both carbon dioxide ($CO_2$) and methane ($CH_4$) to the atmosphere. Global estimates of freshwater emissions traditionally use a wind speed based gas transfer velocity, $k_{CC}$ (introduced by Cole and Caraco (1998)), for calculating diffusive flux with the boundary layer method (BLM). We compared $CH_4$ and $CO_2$ fluxes from BLM with $k_{CC}$ and two other gas transfer velocities ($k_{TE}$ and $k_{HE}$), that include the

5 effects of water-side cooling to the gas transfer besides shear-induced turbulence, with simultaneous eddy covariance (EC) and floating chamber (FC) fluxes during a 16-day measurement campaign in September 2014 at Lake Kuivajärvi in Finland. The measurements included both lake stratification and water column mixing periods. Results show that BLM fluxes were mainly lower than EC, with the more recent model $k_{TE}$ giving the best fit with EC fluxes, whereas FC measurements resulted in higher fluxes than EC. We highly recommend using up to date gas transfer models, instead of $k_{CC}$, for better flux estimates.

BLM $CO_2$ flux had clear diurnal variation with all gas transfer models during both stratified and mixing periods, whereas EC measurements did not detect a diurnal behaviour in $CO_2$ flux. $CH_4$ flux had a diurnal cycle during lake mixing period according to EC and BLM measurements with highest fluxes detected just before sunset. In addition, we found a clear diurnal cycle in the concentration difference between the air and surface water for both $CH_4$ and $CO_2$. This might lead to biased flux estimates, if only daytime values are used in BLM up-scaling and flux measurements in general.

FC measurements did not detect spatial variation in either $CH_4$ or $CO_2$ flux over Lake Kuivajärvi. EC measurements, on the other hand, did not show any spatial variation in $CH_4$ fluxes, but a clear difference between $CO_2$ fluxes from shallower and deeper areas. We highlight that while all flux measurement methods have their pros and cons, it is important to carefully think about the chosen method and measurement interval and their effects on the resulting flux.

# 1 Introduction

Freshwaters (rivers, streams, reservoirs and lakes) are found to be a net source of carbon to the atmosphere (Cole et al., 1994) due to supersaturation of especially carbon dioxide ($CO_2$) but also methane ($CH_4$). Global estimates of the contribution of lakes to the carbon cycle are highly variable and uncertain (Cole et al. (2007); Tranvik et al. (2009); Bastviken et al. (2011); Raymond et al. (2013)), but significant compared to the terrestrial sources and sinks.

Global estimates are usually based on boundary layer method (BLM, also known as boundary layer model) that uses wind speed (via gas transfer velocity $k$) and concentration gradient between the air and surface water as the only factors driving the gas exchange (Cole and Caraco, 1998). According to recent studies, this up-scaling approach strongly underestimates current emissions from lakes and improved methods are needed (e.g. Schubert et al. (2012); Mammarella et al. (2015)). Heiskanen et al. (2014) and Tedford et al. (2014) suggest $k$ models based also on heat flux and water turbulence measurements for more accurate estimates.

A widely used direct flux measurement technique is the floating chamber (FC) method, where the vertical flux at the air-water interface is calculated from the concentration increase within the chamber during the measurement period (Livingston and Hutchinson, 1995). This method has a small source area and is representative of the measurement point only. On the other hand, it can be used to quantify the spatial variability of the gas emissions (Natchimuthu et al., 2016). FC method is laborious, but inexpensive, and does not need extensive data post-processing. However, similar to BLM it requires automatic data loggers or access to a gas analyser, such as gas chromatograph, in the case of manual sampling. FC measurements also disturb the air-water interface and might affect to the gas exchange by creating artificial turbulence, especially with anchored chambers in running waters (Lorke et al., 2015). However, these effects are minor for drifting chambers following the water (Lorke et al., 2015). FC measurements on standing water can also correspond well with non-invasive methods for certain chamber types and deployment methods (Gålfalk et al., 2013).

Recently, also direct eddy covariance (EC) flux measurements have grown their popularity in lake studies, but still there are only few sites with long data sets (e.g. Mammarella et al. (2015), Huotari et al. (2011)). Instead of measuring just a specific point of the lake, the EC method provides flux estimates over a much larger source area, also known as footprint (Aubinet et al., 2012), and as opposed to chamber measurements, it does not disturb the air-water interface. EC measurements are, however, quite expensive and require extensive data post-processing.

In this study, we compared these three flux measurement methods, including three different gas transfer velocities for BLM approach, over a boreal lake in southern Finland for both $CH_4$ and $CO_2$ during an intensive field campaign from 11 September to 26 September in 2014. We also studied spatial variation of $CH_4$ and $CO_2$ fluxes over the EC footprint area with manual floating chambers, while simultaneously estimating fluxes with EC and BLM methods. Our aim is to compare the three methods and make recommendations for future measurements based on our results. Because current up-scaling estimates are based on these methods, comparison is needed to reduce the uncertainties in current estimates of the role of freshwaters in global carbon cycle. Such a comparison also gives valuable information on measurement technique development needs and so far there is only one comparative study including all three methods for $CH_4$ in a temperate lake (Schubert et al., 2012). This is, to our

knowledge, the first study including the three measurement methods for both $CH_4$ and $CO_2$ in a boreal lake, even though the boreal zone harbour a large fraction of the global lakes (Lehner and Döll, 2004; Verpoorter et al., 2014).

## 2 Materials and Methods

### 2.1 Site description and measurements

The study site was the humic, oblong Lake Kuivajärvi situated in southern Finland (61°50' N, 24°17' E), in the middle of a managed mixed coniferous forest, close to the SMEAR II station (Station for Measuring Ecosystem Atmosphere Relations, Hari and Kulmala (2005)). The lake has a maximum depth of 13.2 m, mean depth of 6.3 m, length of 2.6 km and surface area of 0.62 km$^2$ (Fig. 1a). Due to the oblong shape, the wind usually blows along the longest fetch (Mammarella et al., 2015). Lake Kuivajärvi has two separate basins and a measurement raft is mounted on the south basin, near the deepest part of the lake.

Lake Kuivajärvi has median light extinction coefficient $K_d$=0.59 m$^{-1}$ as estimated in Heiskanen et al. (2015). The low water clarity is mainly due to high dissolved organic carbon (DOC) concentration in the lake. Lake Kuivajärvi is a dimictic lake that mixes thoroughly right after ice out usually in the beginning of May, stratifies for summertime and then mixes again latest in October, until it freezes and stratifies again underneath the ice cover for 5–6 months (Heiskanen et al., 2015). These spring and autumn mixing periods usually bring high amounts of $CH_4$ and $CO_2$ from the hypolimnion and bottom sediments of the lake

to the atmosphere (Miettinen et al., 2015).

Continuous measurements of carbon exchange between water and air started already in 2010 and the lake belongs to ICOS (Integrated Carbon Observation System) measuring network. Flux measurement apparatus with the EC system on the raft consists of an ultrasonic anemometer (USA-1, Metek GmgH, Elmshorn, Germany), a closed path infrared gas analyser LI-7200 (LI-COR Inc., Nebraska, USA) for measuring $CO_2$ and water vapour ($H_2O$) mixing ratios and a closed path gas analyser

Picarro G1301-f (Picarro Inc., California, USA) for measuring $CH_4$ and $H_2O$ mixing ratios. Measurement frequency was 10 Hz and a 30-min averaging period was used in this study. $CO_2$ measurements with LI-7200 were stopped on 25 Sept. Air temperature and relative humidity were measured using Rotronic MP102H/HC2-S3 (Rotronic Instrument Corp., NY), while radiation components were measured with Net Radiometer CNR1 (Kipp & Zonen, Delft, Netherlands). These data were collected every 5 s and averaged over 30 min.

Water temperature at depths 0.2, 0.5, 1.0, 1.5, 2.0, 2.5, 3.0, 3.5, 5.0, 6.0, 7.0, 8.0, 10.0 and 12.0 m was measured with a chain of Pt100 temperature sensors. Water column $CO_2$ concentration was measured at depths 0.2, 1.5, 2.5 and 7.0 m using semipermeable silicone tubing in the water and circulating air in a closed loop continuously to the analyser (CARBOCAP®GMP343, Vaisala Oyj, Vantaa, Finland). The measurement system is explained in detail in Hari et al. (2008), Heiskanen et al. (2014) and Mammarella et al. (2015). Water column temperature and $CO_2$ data were collected at the raft every 5 s and averaged over 30

30  min periods.

Another gas analyser (Ultraportable Greenhouse Gas Analyzer, Los Gatos Inc., USA) was used for measuring $CH_4$ and $CO_2$ concentrations in the air at 1 m height and in the water at depths 0.2 and 11 m. The analyser was connected step-wise to three different intakes; one in air, two in water and a dryer, consisting of a container filled with silica gel. For all levels,

air was circulated in closed loop between the gas analyser and the different intakes. The internal pump of the gas analyser was used for this circulation of air at a rate of 1.2 L min$^{-1}$. The air intake consisted of a ca. 10 cm long diffusive membrane (Accurel S6/2, PP, AKZO NOBEL) that was placed under a protective rain cover. The water intakes at each level consisted of a 4.1 m long, 8 mm diameter silicon tube that was bundled and attached to a metal disc ca. 25 cm in diameter, to give a well-defined measurement depth. The dryer was added to the system to remove excess moisture that could have entered into the tubing system by condensation. The air intake was located 1 m above the lake surface and the water intakes were located at 0.2 m and 11 m depths. A full measurement cycle was completed during two hours. The air intake was connected to the gas analyser for 10 min, while the water intakes were connected during 45 minutes each, but data were averaged only during the last 5 minutes of each connection period in order to allow equilibration to the new concentration after a change of intake. After each measurement cycle for the water intakes, the air was circulated through the dryer. The gas analyser was checked against a standard after the measurement campaign and found to be accurate within the specifications of the standard.

Manual floating chamber measurements of $CH_4$ and $CO_2$ fluxes were done with two replicate chambers at eight different spots (Fig. 1b) in the EC footprint area 2–3 times a day (morning, afternoon and night/early morning) during period 11–22 Sept. Unfortunately, measurements were only possible in the first 11 days of the campaign due to high wind and hard weather conditions towards the end. Measurement lines were perpendicular to the shoreline. The line north of the raft was chosen when the wind was blowing from north, and south line was chosen during southerly winds. Measurement spots N2/S2 and N3/S3 were about 10 m deep, and points N1/S1 and N4/S4 were about 3 m deep. They were chosen so that the distance to the measurement raft was about 50 m and the points were marked with buoys.

Chambers used in this study were polyethylene/plexiglas plastic buckets equipped with styrofoam floats and sampling outlets (Gålfalk et al., 2013). Chambers reached approximately 3 cm into the water and their height above water was about 9.6 cm. The closing time for the chambers was 20 min and sampling interval 5 min. Air samples were taken with syringes and injected into 12 ml Labco Exetainer ®vials (Labco Ltd., Lampeter, Ceredigion, UK) and analysed with gas chromatograph (GC). The GC system consisted of a Gilson GX-271 Liquid handler (Gilson Inc., Middleton, USA), a 1 ml Valco 10-port valve (VICI Valco Instrument Co. Inc., Houston, USA) and an Agilent 7890A GC system (Agilent Technologies, Santa Clara, USA) equipped with a flame-ionization detector (temperature 210°C).

In addition to automatic water concentration measurements, we took manual water samples for comparison. Two replicate water samples were taken into 60 ml plastic syringes. After sampling, 30 ml of water was pushed out and replaced by 30 ml of $N_2$ gas. The syringes were placed in a water bath at 20 °C temperature for 30 min. Then the samples were equilibrated by shaking the syringes vigorously for 3 min. The samples of the syringe headspace gas were injected into 12 ml Labco Exetainer ®vials (Labco Ltd., Lampeter, Ceredigion, UK) and analysed with the same GC as manual air samples. Final gas concentrations in the water were calculated using the Henry's Law. Henry's law solubility constants at 298.15 K were for $CH_4$ $1.4 \cdot 10^{-3}$ mol dm$^{-3}$ bar$^{-1}$ (Warneck and Williams, 2012) and for $CO_2$ $3.4 \cdot 10^{-2}$ mol dm$^{-3}$ bar$^{-1}$ (Seinfeld and Pandis, 2016).

## 2.2 Data processing and quality criteria

### 2.2.1 Eddy covariance data

EC data were processed using EddyUH software (Mammarella et al., 2016) according to the approaches in Mammarella et al. (2015). Briefly, spikes in the data were removed on the basis of a maximum difference being allowed between two adjacent points, and 2D coordinate rotation was done so that the wind component $u$ is directed parallel to the mean horizontal wind. Linear detrending was used for calculating the turbulent fluctuations. Lag time was determined from the maximum of the cross-covariance function and cross-wind correction was applied to sonic temperature data (Liu et al., 2001). High frequency spectral corrections were calculated according to Mammarella et al. (2009).

Data quality was ensured with tests for flux stationarity (FST≤1 was approved) and limits for kurtosis (1<$Ku$<8) and skewness (-2<$Sk$<2) (Vickers and Mahrt, 1997). Wind directions other than along the lake were ignored to ensure that only fluxes from the lake were included. Accepted wind directions were 130°<WD<180° and 320°<WD<350°. For gas fluxes, also a criteria for standard deviation of the mixing ratios was used. During night-time, the standard deviation often increased, indicating that there was advection of $CH_4$ and $CO_2$ from the forest uphill to the lake causing scatter in the flux measurements. This scatter was found to be small when the standard deviation of $CO_2$ was less than 3 ppm and thus $CO_2$ mixing ratio (and flux) data with standard deviation larger than 3 ppm were removed. The same procedure was also done for $CH_4$, with the threshold value for standard deviation being 0.003 ppm. After all data quality criteria, the data coverage were 27% and 32% of the original data for $CO_2$ and $CH_4$ fluxes, and 83% and 80% for latent and sensible heat fluxes, respectively. Detection limit for the EC gas fluxes was determined according to Finkelstein and Sims (2001) as $3\sigma$ of the covariance scaled with $\sqrt{N}$, where $N = 48$ was the number of observations per day. This estimate for the detection limit takes into account both instrumental noise and one-point sampling random error (Rannik et al., 2016). On average, detection limit of 30 min averaged $CH_4$ flux was 0.81 nmol m$^{-2}$ s$^{-1}$ and $CO_2$ flux 0.84 $\mu$mol m$^{-2}$ s$^{-1}$. Detection limits scaled for the daily median fluxes were 0.12 nmol m$^{-2}$ s$^{-1}$ and 0.12 $\mu$mol m$^{-2}$ s$^{-1}$ for $CH_4$ and $CO_2$, respectively. The average source area of the EC system reaches 100–300 m from the measurement raft, depending on the stability conditions (Mammarella et al., 2015).

Heat fluxes measured with the EC system were gap-filled using a bulk model depending on water-air temperature difference multiplied by wind speed and vapour pressure difference multiplied by wind speed for sensible and latent heat fluxes, respectively. The coefficients for these relationships were found from a linear fit between measured EC fluxes and the parameters, similar to Mammarella et al. (2015).

### 2.2.2 Chamber flux calculations

The gas concentration increase inside the chambers was linear over a short closure time (20 min) combined with low flux levels. Flux calculation was conducted according to Duc et al. (2013):

$$F = \frac{\mathrm{d}\chi}{\mathrm{d}t} \frac{p_a V}{RTA} \tag{1}$$

where $\frac{dx}{dt}$ is the slope of the linear fit to concentration increase inside the chamber during the closure time ($\mu$l l$^{-1}$s$^{-1}$), $p_a$ ambient pressure (Pa), $V$ chamber volume (m$^3$), $A$ the area of the surface that the chamber covers (m$^2$), $R$ universal gas constant (J mol$^{-1}$K$^{-1}$), and $T$ ambient temperature (K). Measurements were accepted when there were no leakages during the chamber closure. If measurements from both replicate chambers (located within 1 m distance from each other) were successful, then an average flux from these two chambers was used.

## 2.3 Boundary layer method

Diffusive gas exchange $F$ between the air and water was determined according to the boundary layer model

$$F = k(c_{aq} - c_{eq}) \tag{2}$$

where $k$ is the gas transfer velocity (m s$^{-1}$), $c_{aq}$ the gas concentration (mol m$^{-3}$) in surface water and $c_{eq}$ the concentration (mol m$^{-3}$) that the surface water would have if it was in equilibrium with the above air (MacIntyre et al., 1995). Equilibrium gas concentrations were calculated from measurements of mixing ratio $\chi_c$ and air pressure $p_a$ and corrected with Henry's constant $k_H$ according to the solubility of the gas in the water

$$c_{eq} = \chi_c p_a k_H \tag{3}$$

For this study, gas transfer velocity was calculated according to Cole and Caraco (1998), Tedford et al. (2014) and Heiskanen et al. (2014). Gas concentrations for flux calculations were measured both automatically and by manual sampling. Wind speed, sensible and latent heat fluxes and air friction velocity were measured with the EC system.

### 2.3.1 Gas transfer velocity

The most simple and the most often used model for gas transfer velocity $k$ is the one proposed by Cole and Caraco (1998)

$$k_{CC} = (2.07 + 0.215 U_{10}^{1.7}) \left( \frac{Sc}{600} \right)^{-0.5} \tag{4}$$

where $U_{10}$ represents the wind speed at 10 m height (in m s$^{-1}$, approximated by $U_{10} = 1.22U$, where $U$ is the measured wind speed at 1.5 m height) and $Sc$ is the Schmidt number calculated for local conditions. This model considers wind as the only factor causing water turbulence and driving the gas exchange.

A model by Tedford et al. (2014), on the other hand, suggests the importance of the buoyancy flux $\beta$ driven turbulence during cooling periods, so that the turbulent dissipation rate $\varepsilon_{TE}$ becomes

$$\varepsilon_{TE} = \begin{cases} \frac{c_1 u_{*w}^3}{\kappa z} + c_2|\beta| & \text{if } \beta < 0, \\ \frac{c_3 u_{*w}^3}{\kappa z} & \text{if } \beta \geq 0 \end{cases} \tag{5}$$

where $c_1$=0.56, $c_2$=0.77 and $c_3$=0.6 are dimensionless constants, $u_{*w}$ is the friction velocity in the water, $\kappa$=0.41 is the von Karman constant and depth $z$ is here used as constant 0.15 m (Tedford et al. (2014); Mammarella et al. (2015)). Friction

velocity in the water $u_{*w}$ was calculated from direct EC measurements of air friction velocity $u_{*a}$, so that

$$u_{*w} = u_{*a}\sqrt{\frac{\rho_a}{\rho_w}} \tag{6}$$

where $\rho_a$ is the air density and $\rho_w$ water density. Buoyancy flux $\beta$ was calculated according to Imberger (1985):

$$\beta = \frac{g\alpha_t H_{eff}}{\rho_w C_p} \tag{7}$$

where $g$ is the gravitational acceleration, $\alpha_t$ coefficient of thermal expansion of water, $H_{eff}$ the effective heat flux (i.e. latent and sensible heat fluxes and portion of shortwave radiation that is not trapped to the mixing layer are subtracted from the net radiation), and $C_p$ the specific heat of water. Buoyancy flux is positive when the effective heat flux is positive and the lake is heating, whereas negative buoyancy and effective heat fluxes indicate cooling of the lake. Gas transfer velocity $k$ can then be calculated according to the surface renewal model

$$k_{TE} = c_4(\varepsilon_{TE}\nu)^{1/4} Sc^{-1/2}. \tag{8}$$

where $c_4$=0.5 is a dimensionless constant and $\nu$ kinematic viscosity of water (m$^2$ s$^{-1}$).

Another $k$ model that takes heat flux into account as a factor creating turbulence was developed by Heiskanen et al. (2014):

$$k_{HE} = \sqrt{(C_1 U)^2 + (C_2 w_*)^2}\, Sc^{-\frac{1}{2}} \tag{9}$$

Here $C_1 = 0.00015$ and $C_2$=0.07 are dimensionless constants defined for Lake Kuivajärvi (Heiskanen et al., 2014), $w_*$ is the
convective velocity defined as

$$w_* = \sqrt[3]{-\beta z_{AML}} \tag{10}$$

and $z_{AML}$ is the depth of the actively mixing layer (m), where temperature varies within 0.25°C of the surface water temperature. This model was developed in Lake Kuivajärvi for $CO_2$ fluxes but has not been tested for $CH_4$ before this study.

All these three $k$ models are hereafter referred as they are presented in the formulas.

## 3   Results and Discussion

The results of the measurement campaign are divided into two sub-periods (11 days of stratified period 11–22 Sept and 5 days of lake mixing period 22–26 Sept 2014) according to lake stratification and environmental conditions during the campaign, since gas transfer processes differ between these two periods. The water column started its autumn turnover on 22 Sept, but the mixing did not yet reach the lake bottom. Continuous measurements of $CH_4$ and $CO_2$ fluxes with BLM and EC methods are
first compared with each other and the more sporadic FC measurements are then compared to EC measurements by examining the spatial variation and magnitude of the fluxes within the EC footprint area.

## 3.1 Environmental conditions

Weather in the beginning of the measurement campaign in September 2014 was warm with maximum air temperature of 18°C (Fig. 2). Sensible and latent heat fluxes were low, less than 100 W m$^{-2}$ and winds were weak, around 2 m s$^{-1}$ and mostly from south. Air temperature exceeded surface water temperature during the afternoons causing negative sensible heat fluxes. Night-time air temperatures were more than 10°C colder than during daytime.

On 22 Sept, a cold front turned winds north bringing cold air and rain (11 mm on 22 Sept). Air temperature dropped to even 0°C on 24 Sept and wind speeds as high as 8 m s$^{-1}$ were measured at the lake. A drop in the air temperature caused a large temperature difference between air and lake surface water that together with high wind speed caused high, even 200 W m$^{-2}$, positive (upward) sensible and latent heat fluxes on 22 Sept and 23 Sept and a large negative (-400 W m$^{-2}$) effective heat flux, resulting in negative buoyancy flux during this cooling period.

### 3.1.1 Water column temperature and gas profiles

In the beginning of the measurement period, the lake was clearly stratified (Fig. 3a). Bottom temperature was around 9 °C, while surface water temperature was about 16 °C. $CH_4$ concentration according to the automatic measurements at the surface was small, only around 20 nmol m$^{-3}$, while at 11 m depth $CH_4$ concentration was almost 10 times higher than at the surface (Fig. 3b). Manual measurements, on the other hand, show surface water concentrations of 0.07 mmol m$^{-3}$ on average during the stratified period. Manual $CH_4$ concentration measurements were always higher than automatic measurements, which might be caused by insufficient equilibration time for $CH_4$ in the automatic measurement system or by different measurement spots. $CO_2$ concentration at the surface was around 40 mmol m$^{-3}$ according to the automatic measurements and 10 times higher at 11 m depth (Fig. 3c). Manual measurements show $CO_2$ concentration of 110 mmol m$^{-3}$ at the water surface on average. $CO_2$ is more soluble in water than $CH_4$ and thus equilibration time of 40 min should be enough for automatic $CO_2$ measurements. We thereby conclude the difference between automatic and manual $CO_2$ concentration measurements to be caused by spatial variation rather than the measurement system. We point out, however, that choosing the measurement method as well as the measurement spot has an effect on the observed concentrations. Diel variation of $CH_4$ and $CO_2$ concentrations at 11 m could be caused by lake-side cooling and convection or more likely, by internal waves (Stepanenko et al., 2016).

On 14 Sept, the mixing layer of the lake deepened from 5 m to around 6–7 m due to night-time cooling. This mixing brought $CO_2$ rich water from deeper waters to the surface causing a drop in $CO_2$ concentration at 7 m depth and a rapid increase in the surface water concentration according to manual samples. Warm daytime air temperature then caused the surface water to stratify again. Similar occasions of night-time cooling on 16 Sept and 17 Sept induced further decrease in $CO_2$ concentration at 7 m depth and an increase also in the surface water $CO_2$ concentration. After 16 Sept, the automatic and manual $CO_2$ concentration measurements agree better with each other.

On day 22 Sept, a cold front caused the starting of the autumn mixing of the lake. Thermocline reached the depth of 8 m bringing $CH_4$ and $CO_2$ rich water to the surface. Thermocline tilting due to high wind speed caused a rapid increase in 11 m $CH_4$ concentration. $CH_4$ accumulation near the bottom usually happens in the anoxic conditions in late autumn. $CH_4$

concentration at 11 m depth was still three times lower than the maximum concentration found in Stepanenko et al. (2016) in late September. A clear increase in $CH_4$ surface water concentration is seen later, on 23 Sept, and manual measurements show concentrations up to 0.47 mmol m$^{-3}$ on 25 Sept. Decreasing $CO_2$ concentration at 11 m depth on 23-24 Sept was probably due to up-welling. However, this amount of up-welling was not enough to cause a notable increase in the surface water $CO_2$

concentration since $CO_2$ concentration difference between the bottom and the surface is not as drastic as that of $CH_4$, and the gas gets diluted in a large water volume on its way to the surface. Autumn mixing reached 11 m depth in the end of the measurement campaign on 25 Sept, but did not yet mix the bottom waters.

## 3.2 Comparison between boundary layer model and eddy covariance flux estimates

### 3.2.1 $CH_4$ fluxes

$CH_4$ fluxes during the stratified period were small (less than 1 nmol m$^{-2}$s$^{-1}$), estimated both with EC and BLM (Fig. 4). The EC fluxes during the stratified period were close to the detection limit (approximately 0.12 nmol m$^{-2}$ s$^{-1}$ for daily median flux) and are thus highly uncertain. BLM fluxes calculated using the manual surface water concentration measurements were higher than when using automatic measurements, but still small during the stratified period. The difference between manual and automatic BLM fluxes remained below 0.4 nmol m$^{-2}$s$^{-1}$ during the stratified period. After the mixing started on 22 Sept, $CH_4$

fluxes increased rapidly to even 16 nmol m$^{-2}$s$^{-1}$ due to effective mixing and gas transport from deeper waters to the surface. This increase is clearly visible in both EC and BLM fluxes, although BLM flux calculated with $k_{CC}$ remains lower than other BLM fluxes. $CH_4$ flux during the stratified period was considerably lower than 4 nmol m$^{-2}$s$^{-1}$ reported in Miettinen et al. (2015), who used BLM with $k$ calculated from FC measurements, for Lake Kuivajärvi in autumn 2011 and 2012. However, the flux peak in the beginning of the mixing period was over 2-fold to the 6 nmol m$^{-2}$s$^{-1}$ reported in Miettinen et al. (2015),

probably due to rougher weather conditions. Ojala et al. (2011), on the other hand, report high $CH_4$ emissions (6 nmol m$^{-2}$s$^{-1}$) after heavy rain events. Rain on 22 Sept could have also had an effect on lateral $CH_4$ transport from the catchment (Ojala et al. (2011); Rantakari and Kortelainen (2005)). However, in comparison to the situation described by Ojala et al. (2011), the rain episode in Lake Kuivajärvi was very short in duration.

Linear fit parameters for the EC and BLM flux comparison for $CH_4$ show that $k_{TE}$ ($r^2$=0.53) and $k_{HE}$ ($r^2$=0.50) were

similar and comparable to EC measurements, but $k_{CC}$ ($r^2$=0.48) differed from the two others (Table 1). According to the fitting parameters, we can deduce that $k_{TE}$ model gives $CH_4$ fluxes which are almost the same as EC, whereas $k_{CC}$ has the worst agreement with EC measurements (only 50% of the EC measured flux). Ebullition is not an important gas transport mechanism in the EC footprint area as found in Stepanenko et al. (2016) and thus BLM including only diffusive gas flux is expected to give results close to EC. A similar result with $k_{CC}$ giving the lowest flux estimate was also found in Schubert et al.

(2012), where EC and FC methods gave 8 and 7 times higher cumulative fluxes than BLM with $k_{CC}$. Also Blees et al. (2015) report seasonal changes in $CH_4$ flux due to cooling and changes in buoyancy flux. This further encourages to prefer up to date $k$ models instead of $k_{CC}$ in $CH_4$ flux estimates.

### 3.2.2 CO₂ fluxes

$CO_2$ flux was also small (below 1 $\mu$mol m$^{-2}$s$^{-1}$) in the beginning of the measurement campaign due to low wind speeds and thermal stratification of the lake (Fig. 5). Increased surface water concentration in manual samples caused also high BLM flux on 14 and 15 Sept (Fig. 5b). However, this higher flux was not visible in EC measurements or BLM with automatic concentration measurements. On other days, the BLM fluxes calculated using manual samples are slightly higher than the ones calculated using automatic measurements. The difference still remains within 0.2 $\mu$mol m$^{-2}$s$^{-1}$ throughout the measurement period, excluding days 14–15 Sept. The flux increased to almost 3-fold when the lake started mixing with higher wind speeds. Both EC and BLM fluxes show this increase, but $k_{CC}$ model gives clearly lower fluxes than other $k$ models after mixing started. BLM by $k_{TE}$ and $k_{HE}$, on the other hand, agree well with each other during the mixing period. Fluxes before mixing are very similar in magnitude to those reported in Miettinen et al. (2015), Mammarella et al. (2015) and Heiskanen et al. (2014), although the $CO_2$ flux peak measured by BLM with $k_{TE}$ and $k_{HE}$ models in the beginning of the mixing period was larger (3 $\mu$mol m$^{-2}$s$^{-1}$) than reported in other studies from Lake Kuivajärvi (less than 2 $\mu$mol m$^{-2}$s$^{-1}$, Miettinen et al. (2015); Mammarella et al. (2015)). EC, on the other hand, measured daily median $CO_2$ flux less than 2 $\mu$mol m$^{-2}$s$^{-1}$, as reported in other studies. Negative daily median EC flux on 11 Sept and 14 Sept was not statistically different from zero (tested with Mann-Whitney U-test) and denotes a very small flux close to the detection limit of the system (0.12 $\mu$mol m$^{-2}$ s$^{-1}$), rather than uptake which would be very unlikely in September in a boreal lake.

Linear fit parameters for the EC and BLM comparison (Table 1) show that $k_{TE}$ ($r^2$=0.26) and $k_{HE}$ ($r^2$=0.27) give the best results when compared with EC (about 60%). BLM $CO_2$ flux based on $k_{CC}$ was clearly underestimated, being only about 30% of the measured EC flux ($r^2$=0.20). The same result of $k_{CC}$ giving lower fluxes than EC was found also in other studies (e.g. Heiskanen et al. (2014); Mammarella et al. (2015); Podgrajsek et al. (2015)) and the use of this model in global carbon budget estimates may therefore be questionable (e.g. Raymond et al. (2013)). During lake stratification $k_{CC}$ gives the general flux level quite well, while during lake mixing and rain events it is clearly lower than the other modelled fluxes. However, on annual scale, these special occasions might contribute significantly to the $CH_4$ and $CO_2$ budgets (Ojala et al., 2011; Miettinen et al., 2015) and should be noted in up-scaled flux estimates.

Including the effect of lake cooling clearly improves the flux estimate both for $CH_4$ and $CO_2$, albeit these models are not as simple to use as wind speed based models. In the absence of an extensive measurement system, the use of e.g. bulk formulas for estimating latent and sensible heat fluxes for $k_{HE}$ and $k_{TE}$ would result in better flux estimates than the use of $k_{CC}$. Calculating the buoyancy flux (Eq. 7) for $k_{HE}$ and $k_{TE}$ models using bulk formulas for heat fluxes requires an estimate for the depth of the actively mixing layer $z_{AML}$, light extinction coefficient, radiation data, wind speed, as well as temperature and moisture differences between the air and water surface. First, latent and sensible heat fluxes may be calculated from moisture and temperature differences multiplied with wind speed and water vapour or heat transfer coefficients, respectively (Xiao et al., 2013). Net shortwave radiation, $z_{AML}$ and $k_d$ are used to calculate the portion of shortwave radiation that is not trapped to the mixing layer by subtracting entrained shortwave radiation from the radiation remaining at mixing layer depth. With these information, it is possible to calculate the effective heat flux and buoyancy flux, after which estimating $k_{HE}$ and

$k_{TE}$ is straightforward, keeping in mind that the water-side friction velocity for $k_{TE}$ model may be estimated from wind speed measurements by scaling it with an appropriate drag coefficient.

## 3.3 Diurnal variation of estimated fluxes

In order to deepen the comparison between the methods, diurnal variation of $CH_4$ and $CO_2$ fluxes are analysed for the two
study periods separately. Diurnal variation of $CH_4$ flux during the stratified period was negligible (results not shown), but $CO_2$ flux variation was separately studied for the two periods: stratified and lake mixing periods. The sun rose at 5:45 and set at 18:45 during the stratified period whereas during the mixing period sunrise was at 6:15 and sunset at 18:15.

### 3.3.1 Stratified period

BLM $CO_2$ fluxes had clear diurnal variation before mixing (Fig. 6). BLM fluxes by $k_{HE}$ and $k_{CC}$ show similar diurnal pattern
with lowest flux in late afternoon, although $k_{CC}$ results in a remarkably lower flux than $k_{HE}$ in general. Low BLM fluxes in the daytime ($0.305\pm0.009$ and $0.201\pm0.004$ $\mu$mol m$^{-2}$s$^{-1}$ on average with $k_{HE}$ and $k_{CC}$ models, respectively) are probably caused by photosynthetic activity of algae in the lake that reduces the $CO_2$ concentration difference between air and water ($\Delta[CO_2]$) right after sunrise (Fig. 7d, Table 2). Also the convective term ($C_2 w_*$) in $k_{HE}$ is negligible during daytime when the lake is heating due to higher air temperature, resulting in a lower $k_{HE}$ (Fig. 7a). Higher flux during night-time ($0.410\pm0.008$
on average with $k_{HE}$ model) is probably caused by turbulence created by waterside cooling. This is seen in Fig. 7a as the convective term in $k_{HE}$ increases towards night-time causing higher total $k_{HE}$. Podgrajsek et al. (2015) argued that the main driver for enhanced night-time gas exchange is convection, and they did not find a correlation with the concentration difference $\Delta[CO_2]$. However, we find that also $\Delta[CO_2]$ increases during night-time in the lack of algal photosynthesis. The magnitude of the BLM fluxes with $k_{HE}$ and $k_{CC}$ are, however, quite different, and $k_{CC}$ gives lower fluxes throughout the day and no clear
difference in average daytime and night-time fluxes (Figs. 6a and 6b, Table 2). $CO_2$ flux is especially underestimated during night-time by $k_{CC}$, when night-time cooling and convective mixing are more important, because it lacks the convective term.

BLM by $k_{TE}$ gives highest fluxes at noon when also friction velocity gains its maximum value (Fig. 7c). The BLM flux by $k_{TE}$ is thus also larger in the daytime ($0.545\pm0.014$ $\mu$mol m$^{-2}$s$^{-1}$ on average, Table 2) despite the lower $\Delta[CO_2]$, and night-time flux ($0.396\pm0.010$ $\mu$mol m$^{-2}$s$^{-1}$) is 27% smaller than the daytime flux. Water friction velocity, that was used in $k_{TE}$,
was calculated from direct EC measurements in the air (Eq. 6). Friction velocity calculated from wind speed measurements (with a drag coefficient 0.001 for a water surface) instead of direct $u_{*a}$ measurements gave similar diurnal variation as models $k_{HE}$ and $k_{CC}$ (data not shown), but resulted in a lower $u_{*w}$. BLM with $k_{TE}$ could give better results with direct turbulence measurements in the water. The buoyancy term ($\beta$) in $k_{TE}$ is low compared to the shear term ($u_*^3/(\kappa z)$) throughout the day (Fig. 7c). EC flux does not show any diurnal variation for $CO_2$ exchange over the lake when the lake is stratified (Fig. 6d).
Vesala et al. (2006) found the same result in $CO_2$ EC flux in September over a small humic lake in Finland with fluxes usually under 1 $\mu$mol m$^{-2}$s$^{-1}$ during the stratified period. Overall, $k_{HE}$ and EC measurements agree well on the magnitude of $CO_2$ flux during daytime, but night-time values differ.

### 3.3.2 Mixing period

During the mixing period all BLM as well as EC fluxes show similar diurnal pattern in $CH_4$ flux, so that the highest flux value is reached in the afternoon/evening, just before sunset (Fig. 8). EC measurements, however, miss the early morning flux peak detected with BLM models just before sunrise. Because the afternoon flux peak is also seen in the BLM by $k_{CC}$, we can deduce that it is due to higher wind speed and enhanced shear during the afternoon as well as higher $CH_4$ concentration difference ($\Delta[CH_4]$) between the surface water and air, that is also partly due to enhanced mixing bringing $CH_4$ from deeper waters (Fig. 9d). The larger concentration difference $\Delta[CH_4]$ towards the afternoon may be caused by higher oxidation rate in dark that lowers $CH_4$ concentration in the water during night (Mitchell et al., 2005). During daytime solar radiation, the oxidation rate would then be lower resulting in an increase of water $CH_4$ concentration towards the afternoon. Another possibility for larger concentration difference $\Delta[CH_4]$ in the afternoon, in addition to $CH_4$ feeding from the deeper waters and lower oxidation rate, is enhanced resuspension from the sediments in the littoral zone during periods of high wind speed (Bussmann, 2005). Rain on 22 Sept could have also enhanced transport from the catchment to the lake (Ojala et al., 2011). EC and BLM fluxes by $k_{HE}$ and $k_{TE}$ are also similar in magnitude ($5.9\pm0.3$, $7.1\pm0.6$ and $7.7\pm0.6$ nmol m$^{-2}$s$^{-1}$ daytime averages, respectively), whereas $k_{CC}$ gives clearly lower fluxes ($3.7\pm0.3$ nmol m$^{-2}$s$^{-1}$ daytime average, Table 2). All the models give similar diurnal patterns of $CH_4$ flux, only the magnitudes are different. Night-time minimum flux values were 90%, 95% and 91% smaller than the daytime maximum for $k_{HE}$, $k_{CC}$ and $k_{TE}$ fluxes, respectively. Models $k_{HE}$ and $k_{TE}$ show $CH_4$ flux variation quite similar to EC also in magnitude (Table 2). Keller and Stallard (1994), Bastviken et al. (2004) and Bastviken et al. (2010) also report highest daytime fluxes for $CH_4$ probably caused by more effective turbulent transfer during daytime, while Podgrajsek et al. (2014b) report higher night-time fluxes and suggest it to be caused by water-side convection. However, we find that both surface water concentration changes and more effective daytime gas transfer are likely explanations to the higher daytime $CH_4$ fluxes in Lake Kuivajärvi.

After mixing started, all models agreed well on diurnal variation of $CO_2$ flux with higher fluxes during daytime and lower during night (Fig. 10). Average daytime $CO_2$ fluxes were $1.3\pm0.2$, $2.15\pm0.06$, $2.37\pm0.06$ and $1.11\pm0.04$ $\mu$mol m$^{-2}$s$^{-1}$ with EC method and BLM by $k_{HE}$, $k_{TE}$ and $k_{CC}$, respectively and night-time average fluxes notably smaller, as $0.88\pm0.14$, $1.43\pm0.05$, $1.54\pm0.05$ and $0.58\pm0.02$ $\mu$mol m$^{-2}$s$^{-1}$ with EC method and BLM by $k_{HE}$, $k_{TE}$ and $k_{CC}$, respectively (Table 2). Night-time lowest fluxes were 60%, 76% and 68% lower than the daytime maximum BLM fluxes with $k_{HE}$, $k_{CC}$ and $k_{TE}$ models, respectively. Highest flux according to BLM with all three $k$ models is gained at noon when wind speeds are highest, even though $\Delta[CO_2]$ is at minimum (Fig. 9d). Shear terms $C_1U$ and $u_*^3/(\kappa z)$ in $k_{HE}$ and $k_{TE}$ models, respectively, have diurnal variations with highest values at noon as well (Figs. 9a and 9c), which is then visible in the diurnal variations of fluxes (Figs. 10a and 10c). BLM by $k_{CC}$, however, shows considerably lower fluxes than $k_{HE}$ and $k_{TE}$ both during daytime and night-time (Fig. 10b, Table 2). Higher fluxes during daytime than night-time in the mixing period are expected due to enhanced gas transfer during stronger winds in the daytime. The buoyancy term $\beta$ in $k_{TE}$ is still almost a magnitude smaller than the shear term and does not influence the $k_{TE}$ much, even during lake mixing (Fig. 9c).

The maximum and minimum concentration differences $\Delta[CO_2]$ were 1.4 to 1.6 times higher during the mixing period than in the stratified period. This may be caused by up-welling of $CO_2$ from deep waters to the surface and algal photosynthesis at the surface. This indicates that using selectively only daytime gas concentration measurements in flux measurements and global budgets already makes a biased assumption. The EC measured $CO_2$ flux does not show a clear diurnal variation during this period either.

## 3.4 Comparison between floating chambers and eddy covariance fluxes

In addition to comparison between FC and EC measurements, spatial variation of $CH_4$ and $CO_2$ fluxes within the EC footprint area was also studied with floating chambers at different parts of the lake during the stratified period 11–22 Sept 2014. The measurement spots were chosen upwind from the measurement raft to ensure being within the EC footprint area. Results are shown in Fig. 11, where the median of FC measurements at different spots are compared with the median of simultaneous EC measurements.

### 3.4.1 CH$_4$ fluxes

During the stratified period, $CH_4$ fluxes measured with the FC method were very small, mainly less than 2 nmol m$^{-2}$s$^{-1}$ (Fig. 11a). The average of all FC $CH_4$ flux measurements was 1.67 nmol m$^{-2}$s$^{-1}$ and the coefficient of variation was 1.25. FC $CH_4$ fluxes were systematically higher than EC fluxes (statistical significance tested with Mann-Whitney U-test, $p < 0.01$), as also observed in Eugster et al. (2011). Daytime average FC $CH_4$ flux was 2.4±0.3 nmol m$^{-2}$s$^{-1}$ whereas daytime EC flux was only 0.41±0.04 nmol m$^{-2}$s$^{-1}$. Night-time average FC $CH_4$ flux was 1.1±0.2 nmol m$^{-2}$s$^{-1}$ and EC flux 0.34±0.04 nmol m$^{-2}$s$^{-1}$ (Table 2). There is a clear difference between these methods during both day and night, although daytime difference is more remarkable. Partly this difference is of course due to FC fluxes averaged over the different measurement spots, and measurement points N3 and N4 showed slightly higher FC fluxes than elsewhere.

Other possible reason for the difference could be that the chambers were anchored to the boat during flux measurements, which might create artificial turbulence, although Gålfalk et al. (2013) did not find a significant difference between anchored and drifting chambers with this particular chamber design. A more probable reason behind the result is that these low fluxes are very difficult to detect with the EC method, since the $CH_4$ fluxes were very close to the detection limit of the gas analyser used in EC measurements. Higher fluxes during the mixing period could have probably produced a better comparison. Podgrajsek et al. (2014a) did not find systematically higher fluxes with EC or FC and found quite good agreement between these two methods for $CH_4$ fluxes. In this study EC and FC $CH_4$ fluxes did not compare well with each other and the difference in fluxes is statistically significant, mainly due to too low $CH_4$ fluxes for the EC method to detect reliably. EC method has a larger source area than FC method, which might also affect the flux. Windy conditions during the mixing period could have made the comparison better, but manual FC measurements are difficult to do during high wind and rough weather conditions.

### 3.4.2 CO$_2$ fluxes

During the stratified period, CO$_2$ flux varied around 0.2–0.6 $\mu$mol m$^{-2}$s$^{-1}$ when measured with FC, whereas EC measured fluxes varied between 0.3–0.4 $\mu$mol m$^{-2}$s$^{-1}$ (Fig. 11). The average FC CO$_2$ flux was 0.40 $\mu$mol m$^{-2}$s$^{-1}$ and the coefficient of variation was 0.63 (Fig. 11b). Daytime average FC CO$_2$ flux was 0.62$\pm$0.08 $\mu$mol m$^{-2}$s$^{-1}$ and differed from daytime EC
5 CO$_2$ flux (0.31$\pm$0.04 $\mu$mol m$^{-2}$s$^{-1}$). Night-time fluxes, however, are not different between FC and EC methods (0.29$\pm$0.04 $\mu$mol m$^{-2}$s$^{-1}$ when measured with FC and 0.28$\pm$0.08 $\mu$mol m$^{-2}$s$^{-1}$ with EC, Table 2). CO$_2$ fluxes were almost always higher when measured with FC than EC method (statistical significance tested with Mann-Whitney U-test, $p < 0.01$) and the FC measurements did not show spatial variation. Eugster et al. (2003) also report higher CO$_2$ flux when measured with FC compared to EC. However, from the north side of the measurement raft (measurement spots N1–N4), FC fluxes do not differ
10 statistically from EC CO$_2$ fluxes.

 There is a clear difference between EC measurements from the south and north sides of the lake (tested with Mann-Whitney U-test, $p < 0.01$) with approximately 0.1 $\mu$mol m$^{-2}$s$^{-1}$ higher CO$_2$ fluxes measured from the south than from north (Fig. 11b). South side of the raft is shallower than the north side (Fig. 1a) and thus more prone for the mixing to reach bottom even during the stratified period. The EC footprint area of 100–300 m (Mammarella et al., 2015) from the raft reaches further to the shallow
15 areas than the FC measurements that were done approximately 50 m south from the raft. EC is thus more likely to catch the higher gas fluxes resulting from up-welling of gas-rich waters from the bottom. Higher CH$_4$ flux from the south side is not detected probably due to CH$_4$ oxidation in the water column into CO$_2$. This oxidation would not increase the CO$_2$ efflux, as CH$_4$ flux is so much smaller than that of CO$_2$. Footprint area north from the raft is over significantly deeper water and mixing from the deeper waters during stratified period is unlikely.

20 EC measurement systems are set up in one place, often on the shore or on a raft near the deepest parts of the lake to have a large footprint area for measurements. This is due to limitations in the EC method, because it requires a homogeneous surface and favourable wind conditions, but leads to possibly biased flux estimations, especially if flux is only measured over a particularly deep or shallow area. FC method is good for detecting spatial variation, but has its limitations regarding temporal and spatial data coverage and challenging measurement conditions.

### 4 Conclusions

We found that all gas transfer velocity, $k$, models used in BLM calculation gave mainly lower flux estimates of both CH$_4$ and CO$_2$ compared to EC, while FC measurements were higher than EC. For CH$_4$ fluxes, this difference between FC and EC methods is probably caused by the EC system detection limit that was very close to the measured fluxes during lake stratification. For CO$_2$, there was no statistical difference between FC and EC methods over the north side of the lake and
30 night-time average fluxes were almost the same with these two methods. Gas transfer velocity models by Tedford et al. (2014) ($k_{TE}$) and Heiskanen et al. (2014) ($k_{HE}$) showed very similar fluxes both for CH$_4$ and CO$_2$, and the $k$ model by Cole and Caraco (1998) ($k_{CC}$) resulted in clearly lower gas fluxes especially during the lake mixing period. A comparison between BLM and EC fluxes showed that, on average, the $k_{TE}$ model is the most similar and the $k_{CC}$ model the lowest, when compared to EC

fluxes. For global up-scaling, it would be preferable to use up to date $k$ models instead of $k_{CC}$ to reduce the risk of systematic biases. The simple $k_{CC}$ model underestimates the flux especially during special occasions of e.g. lake mixing and rain events, which may vastly contribute to the annual flux estimate.

Diurnal variation of $CH_4$ and $CO_2$ fluxes was examined by BLM and EC measurements. During the mixing period the BLM with different $k$ models agreed well with each other on the shape of the variation both for $CH_4$ and $CO_2$ fluxes, but the magnitudes differed between the models. During the stratified period, $CO_2$ flux by $k_{TE}$ showed an opposite diurnal pattern than other models due to higher air friction velocity during daytime. This model could work better with direct friction velocity measurements in the water. The buoyancy term included in $k_{TE}$ model was not significant compared to the shear term and does not affect the diurnal variation of the flux. $CO_2$ concentration difference between the surface water and air was found to have a diurnal cycle with lower values during daytime, probably due to algal photosynthesis reducing surface water concentration of $CO_2$. An opposite diurnal cycle was found for $CH_4$ concentration difference with highest values reached in the afternoon. This might be due to $CH_4$ feeding from the deeper waters, lower oxidation rate in daylight in the water column, or due to more effective lateral transport from the littoral zone during higher wind speeds in the daytime. As we observe a clear diurnal cycle in the concentration difference for both $CH_4$ and $CO_2$, it is important to note that using only daytime concentration (and wind speed) measurements for up-scaling with BLM affects the resulting flux estimate.

FC measurements did not show a spatial variation in either $CH_4$ or $CO_2$ flux. $CO_2$ EC flux was clearly higher from the south side of the measurement raft than north, due to shallower lake area within the EC footprint on the south side. This was not detected with $CH_4$, probably due to oxidation in the water column.

FC measurements are generally used for studying spatial variation, but our results suggest that also EC measurements are able to detect differences between different wind sectors. As we find clear differences between night-time and daytime flux measurements as well as between stratified and lake mixing periods, it is advisable to prefer frequent and diverse sampling over daytime-only measurements, that can lead to biases in greenhouse gas budget estimates.

## 5    Data availability

Eddy covariance, water column temperature and $CO_2$ concentration and meteorological data are available in AVAA - Open research data publishing platform (http://openscience.fi/avaa). The metadata of the observations are available via ETSIN–service. Data from manual measurements are available upon request from the first author.

*Author contributions.*  IM, DB, JH, MR and TV designed the field experiments. KME, MR, AO and JH carried out manual field measurements. KME, IM and OP participated in eddy covariance data processing and analysis. TB and AL carried out automatic gas concentration measurements in the water column. All authors participated in analysing the results and KME prepared the manuscript with contributions from all co-authors.

*Competing interests.*  The authors declare that they have no conflict of interest.

*Acknowledgements.* We thank the Hyytiälä Forestry Field Station staff for all their technical support and Maria Gutierrez de los Rios for her help during the measurement campaign. This study was supported by EU project GHG-LAKE (612642), Academy of Finland (CarLAC (281196) project, Centre of Excellence (272041), Academy Professor projects (284701 and 282842)), SRC-VR and ERC 725546 and ICOS-FINLAND (281255).

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

**Table 1.** Linear fit $y = ax + b$ parameters for comparison between EC and BLM fluxes according to different models for $k$, when EC flux estimates were on the x-axis. Uncertainties are given by the standard errors of the parameters. The comparison was made using daily median fluxes calculated from 1/2 h averages.

| | Model | a | b [nmol m$^{-2}$s$^{-1}$] | $r^2$ | RMSE [nmol m$^{-2}$s$^{-1}$] |
|---|---|---|---|---|---|
| | $k_{HE}$ | 0.9±0.2 | -0.3±0.8 | 0.50 | 2.62 |
| CH$_4$ | $k_{TE}$ | 1.0±0.2 | -0.3±0.8 | 0.53 | 2.58 |
| | $k_{CC}$ | 0.5±0.1 | -0.2±0.4 | 0.48 | 1.38 |
| | Model | a | b [$\mu$mol m$^{-2}$s$^{-1}$] | $r^2$ | RMSE [$\mu$mol m$^{-2}$s$^{-1}$] |
| | $k_{HE}$ | 0.6±0.3 | 0.3±0.2 | 0.27 | 0.58 |
| CO$_2$ | $k_{TE}$ | 0.6±0.3 | 0.4±0.2 | 0.26 | 0.59 |
| | $k_{CC}$ | 0.3±0.1 | 0.2±0.1 | 0.20 | 0.30 |

**Table 2.** Average daytime and night-time $CH_4$ and $CO_2$ fluxes during lake stratification and mixing periods using different measurement methods. Note that FC fluxes are averaged also over different measurement spots. Mixing period did not include enough FC measurements for this analysis. Uncertainties are given by the standard errors of the flux averages.

| Stratified period | $CH_4$ flux [nmol m$^{-2}$s$^{-1}$] | | $CO_2$ flux [$\mu$mol m$^{-2}$s$^{-1}$] | |
|---|---|---|---|---|
| | day | night | day | night |
| BLM $k_{HE}$ | 0.177 ($\pm$ 0.005) | 0.431 ($\pm$ 0.008) | 0.305 ($\pm$ 0.009) | 0.410 ($\pm$ 0.008) |
| BLM $k_{TE}$ | 0.370 ($\pm$ 0.011) | 0.439 ($\pm$ 0.007) | 0.545 ($\pm$ 0.014) | 0.396 ($\pm$ 0.010) |
| BLM $k_{CC}$ | 0.128 ($\pm$ 0.003) | 0.186 ($\pm$ 0.004) | 0.201 ($\pm$ 0.004) | 0.180 ($\pm$ 0.004) |
| EC | 0.41 ($\pm$ 0.04) | 0.34 ($\pm$ 0.04) | 0.31 ($\pm$ 0.04) | 0.28 ($\pm$ 0.08) |
| FC | 2.4 ($\pm$ 0.3) | 1.1 ($\pm$ 0.2) | 0.62 ($\pm$ 0.08) | 0.29 ($\pm$ 0.04) |
| Mixing period | $CH_4$ flux [nmol m$^{-2}$s$^{-1}$] | | $CO_2$ flux [$\mu$mol m$^{-2}$s$^{-1}$] | |
| | day | night | day | night |
| BLM $k_{HE}$ | 7.1 ($\pm$ 0.6) | 6.6 ($\pm$ 0.5) | 2.15 ($\pm$ 0.06) | 1.43 ($\pm$ 0.05) |
| BLM $k_{TE}$ | 7.7 ($\pm$ 0.6) | 7.1 ($\pm$ 0.5) | 2.37 ($\pm$ 0.06) | 1.54 ($\pm$ 0.05) |
| BLM $k_{CC}$ | 3.7 ($\pm$ 0.3) | 2.8 ($\pm$ 0.2) | 1.11 ($\pm$ 0.04) | 0.58 ($\pm$ 0.02) |
| EC | 5.9 ($\pm$ 0.3) | 5.0 ($\pm$ 0.4) | 1.3 ($\pm$ 0.2) | 0.88 ($\pm$ 0.14) |

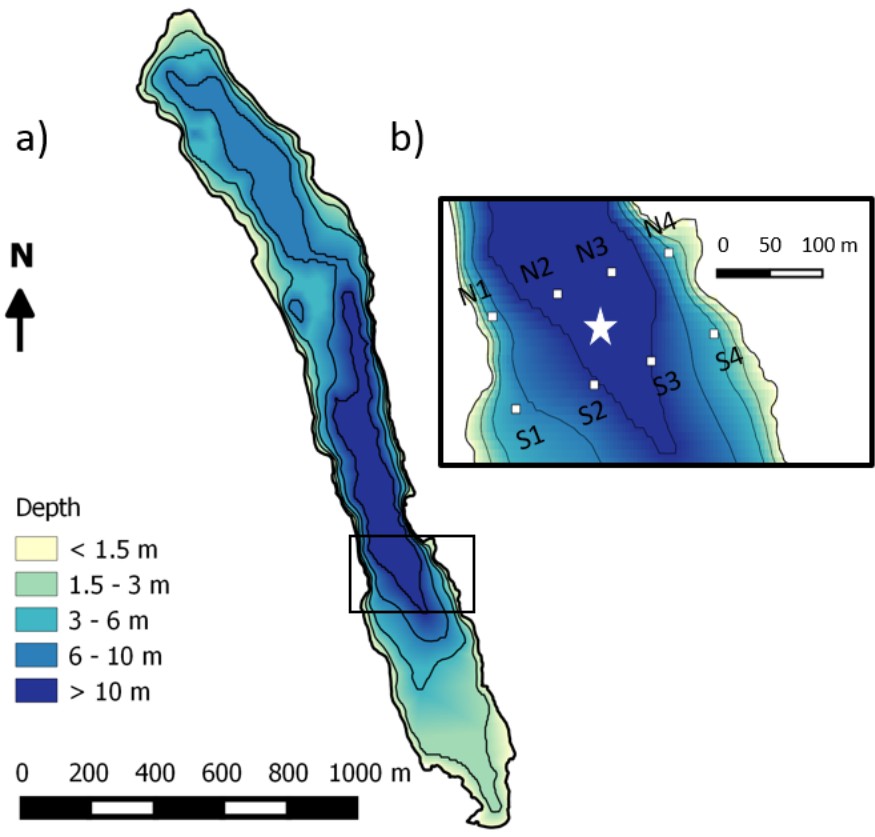

**Figure 1.** (a) Bathymetry of Lake Kuivajärvi and (b) floating chamber measurement spots (white squares) around the EC measurement raft (white star).

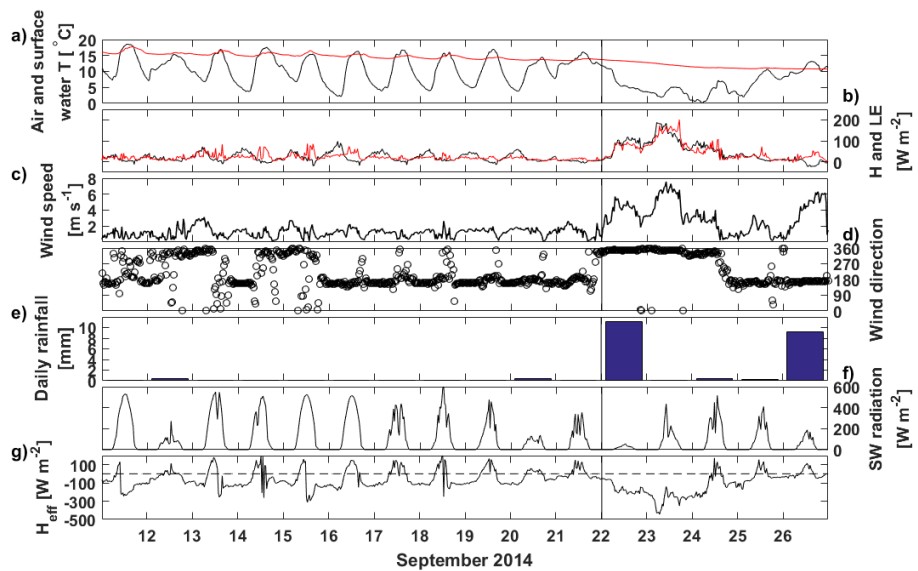

**Figure 2.** Half hour averages of (a) measured air temperature (black) and lake surface water temperature (red), (b) sensible (black) and latent (red) heat fluxes measured with the EC system and gap-filled using a bulk formula (see Sect. 2.2.1 and Mammarella et al. (2015) for details), (c) wind speed, (d) wind direction, (e) daily rainfall, (f) incoming shortwave radiation and (g) effective heat flux measured at the measurement raft. Time ticks represent midnight and the vertical black line the start of the lake mixing period.

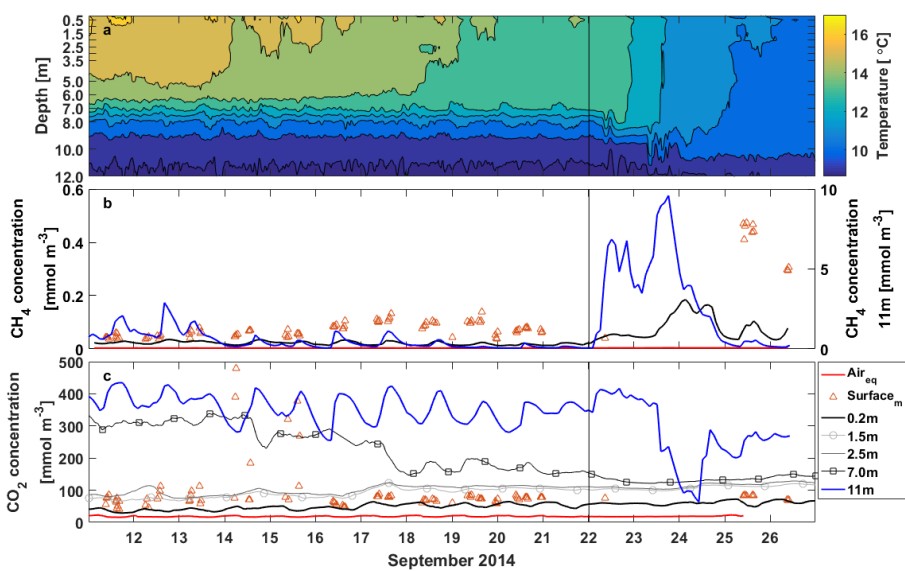

**Figure 3.** Half hour averages of (a) temperature, (b) $CH_4$ concentration and (c) $CO_2$ concentration in the water column at different depths. The red line is the equilibrium concentration of $CH_4$ and $CO_2$ at the surface in subplots b and c, respectively. The orange triangles are manual headspace samples taken from the surface water at chamber measurement locations. Time ticks represent midnight and the vertical black line the start of the lake mixing period. Note that $CH_4$ concentration at 11 m depth (blue line) is read from the right y-axis.

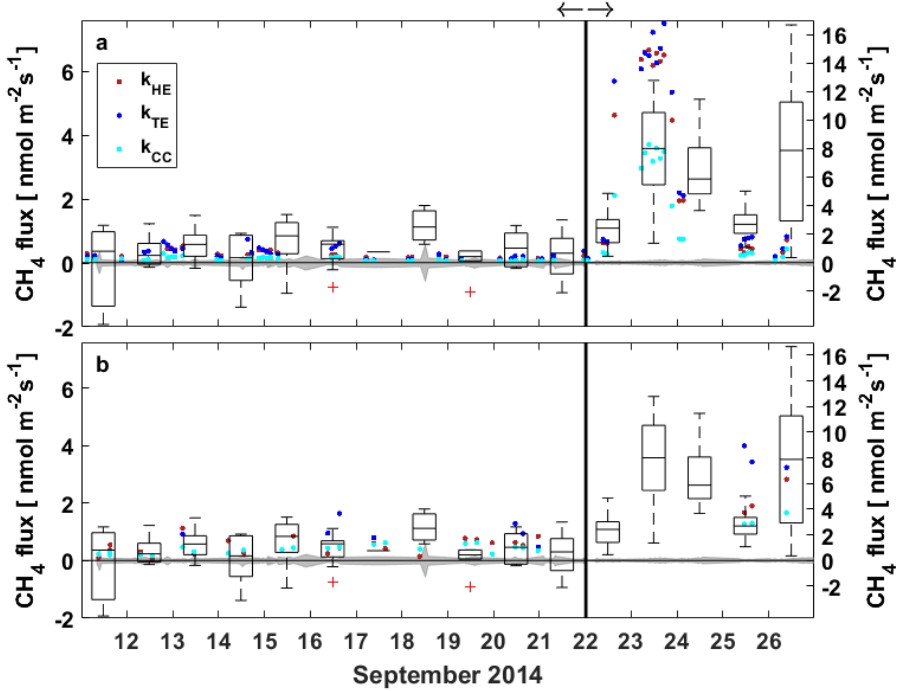

**Figure 4.** Daily median EC CH$_4$ flux (boxplot) and BLM CH$_4$ fluxes with different models for gas transfer velocity $k$ using surface water CH$_4$ concentration from (a) automatic measurements and (b) manual water samples. In the boxes, the middle black line is the daily median flux while the bottom and top edges of the boxes indicate the 25th and 75th percentiles, respectively. The black whiskers show the most extreme data points that are not considered as outliers ($\pm 2.7\sigma$) and the outliers are represented with the red '+' symbol. Gray shadowed area shows the flux detection limit ($3\sigma$ of the covariance). Time ticks represent midnight and the vertical black line the start of the lake mixing period. Fluxes during the stratified period (11-22 Sept) are read from the left and mixing period fluxes (22-26 Sept) from the right axis.

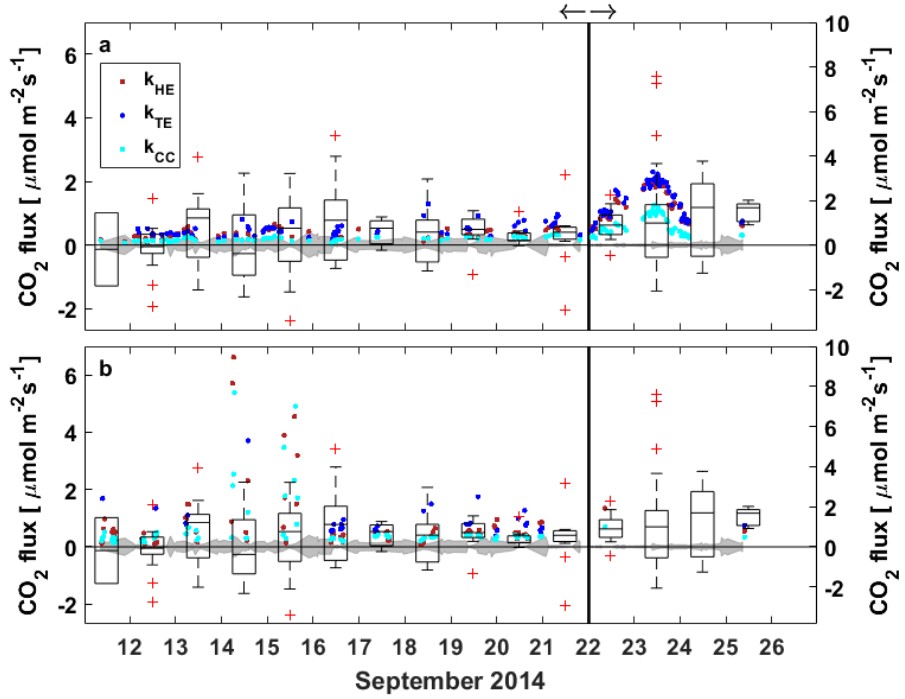

**Figure 5.** Daily median EC $CO_2$ flux (boxplot) and BLM $CO_2$ fluxes with different models for gas transfer velocity $k$ using surface water $CO_2$ concentration from (a) automatic measurements and (b) manual water samples. In the boxes, the middle black line is the daily median flux while the bottom and top edges of the boxes indicate the 25th and 75th percentiles, respectively. The black whiskers show the most extreme data points that are not considered as outliers ($\pm 2.7\sigma$), and the outliers are represented with the red '+' symbol. Gray shadowed area shows the flux detection limit ($3\sigma$ of the covariance). Time ticks represent midnight and the vertical black line the start of the lake mixing period. Fluxes during the stratified period (11-22 Sept) are read from the left and mixing period fluxes (22-26 Sept) from the right axis.

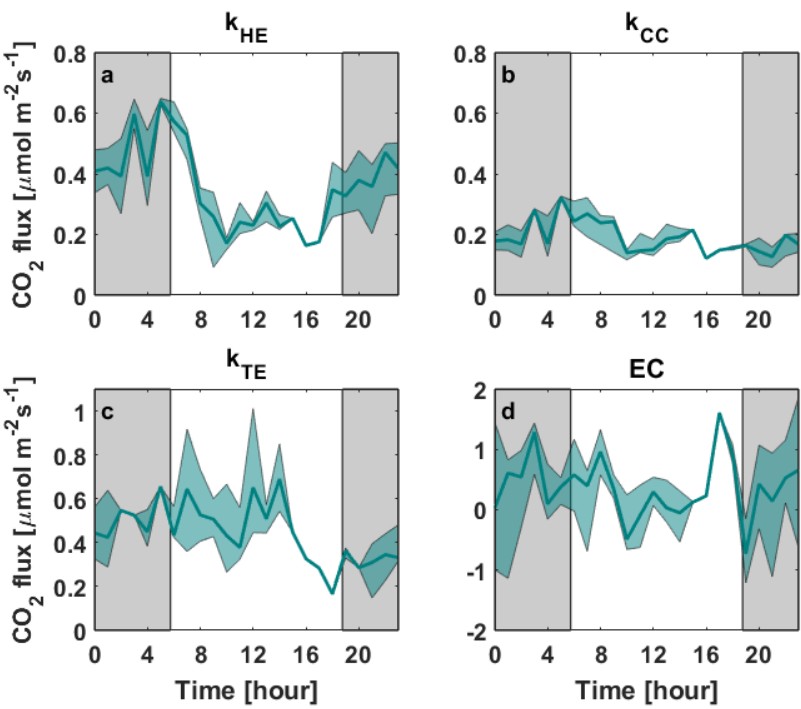

**Figure 6.** Diurnal variation of $CO_2$ flux ($\mu$mol m$^{-2}$s$^{-1}$) using the BLM method with gas transfer velocity calculated according to (a) $k_{HE}$, (b) $k_{CC}$ and (c) $k_{TE}$, and (d) the measured EC flux during the stratified period 11 –22 September 2014. Thick blue line is the median value and the light blue area shows the 25th and 75th percentiles. Gray areas represent night-time. Note the change in y-axis scale in subplots c and d.

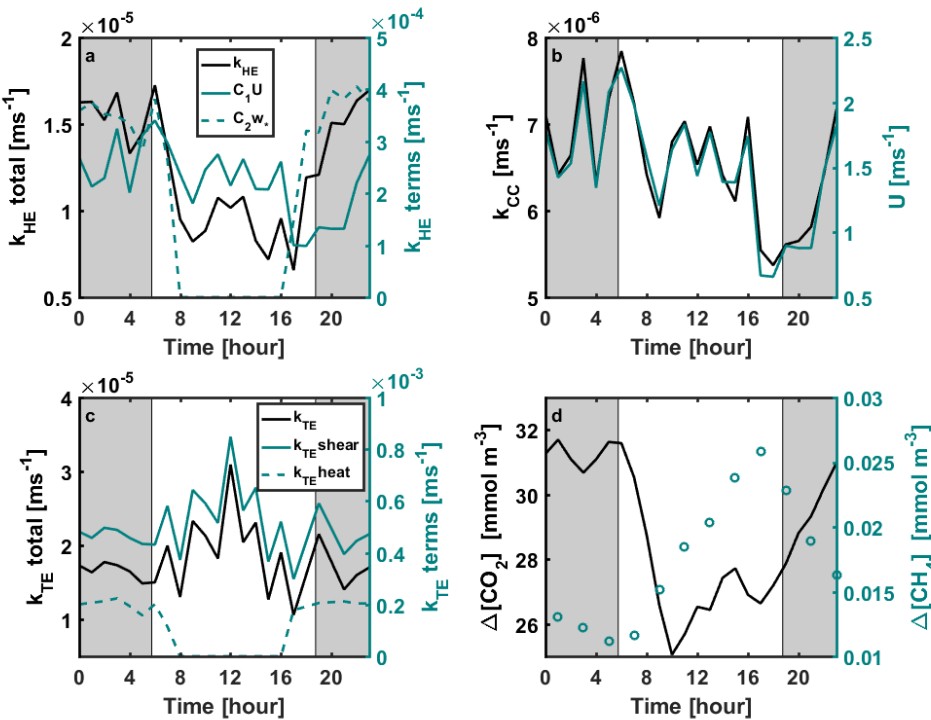

**Figure 7.** Diurnal variation of (a) $k_{HE}$ and its shear and convective terms (Eq. 9), (b) $k_{CC}$ and wind speed, (c) $k_{TE}$ and its shear ($k_{TE}shear = \frac{c_1 u_{*w}^3}{\kappa z}$ or $k_{TE}shear = \frac{c_3 u_{*w}^3}{\kappa z}$) and convective ($k_{TE}heat = c_2|\beta|$ or $k_{TE}heat = 0$) terms (Eq. 8) and (d) $CO_2$ and $CH_4$ concentration differences between air and surface water during the stratified period 11–22 September 2014. Shear and convective terms in subplots a and c are not corrected with the Schmidt number. Gray areas represent night-time.

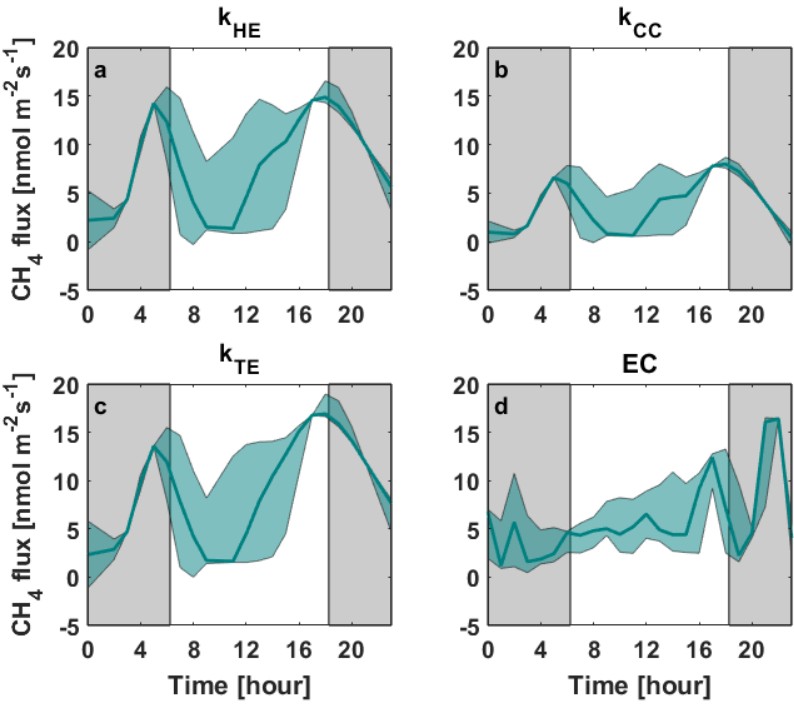

**Figure 8.** Diurnal variation of $CH_4$ flux (nmol m$^{-2}$s$^{-1}$) using the BLM method with gas transfer velocity calculated according to (a) $k_{HE}$, (b) $k_{CC}$ and (c) $k_{TE}$, and (d) the measured EC flux during the mixing period 22 –26 September 2014. Thick blue line is the median value and the light blue area shows the 25th and 75th percentiles. Gray areas represent night-time.

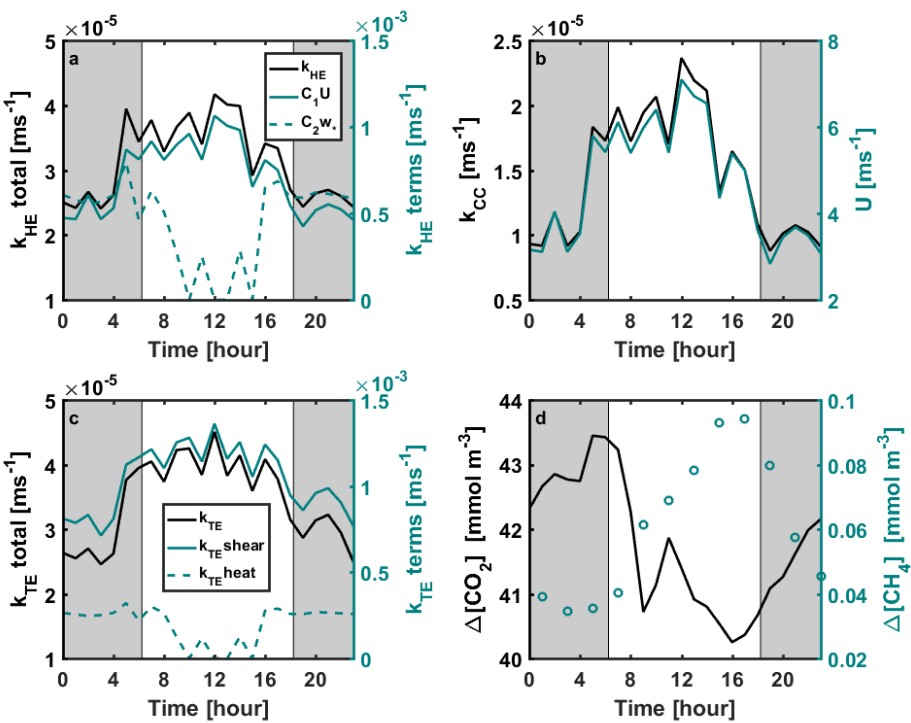

**Figure 9.** Diurnal variation of (a) $k_{HE}$ and its shear and convective terms, (b) $k_{CC}$ and wind speed, (c) $k_{TE}$ and its shear ($k_{TE}shear = \frac{c_1 u_{*w}^3}{\kappa z}$ or $k_{TE}shear = \frac{c_3 u_{*w}^3}{\kappa z}$) and convective ($k_{TE}heat = c_2|\beta|$ or $k_{TE}heat = 0$) terms (Eq. 8) and (d) $CO_2$ and $CH_4$ concentration differences between air and surface water during the mixing period 22–26 September 2014. Shear and convective terms in subplots a and c are not corrected with the Schmidt number. Gray areas represent night-time.

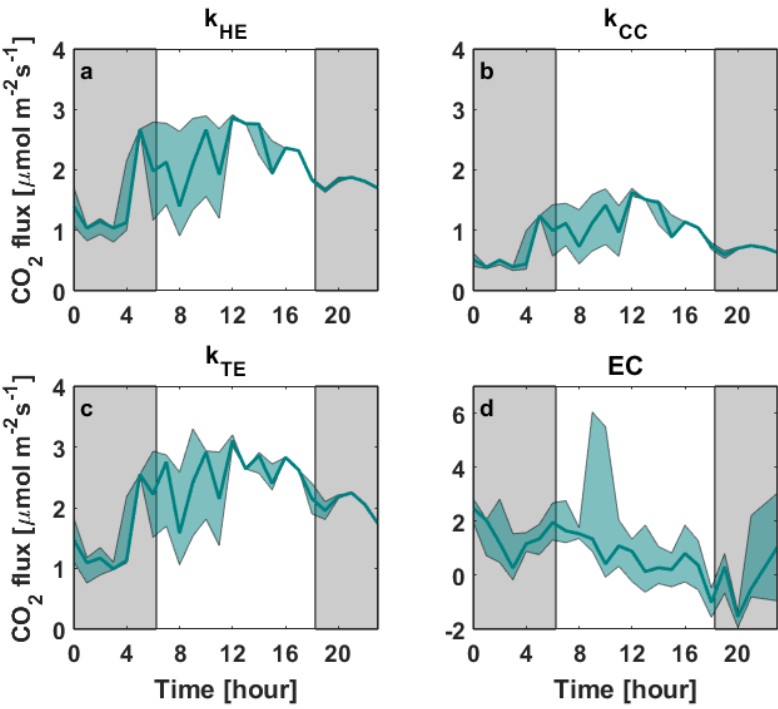

**Figure 10.** Diurnal variation of $CO_2$ flux ($\mu$mol m$^{-2}$s$^{-1}$) using the BLM method with gas transfer velocity calculated according to (a) $k_{HE}$, (b) $k_{CC}$ and (c) $k_{TE}$, and (d) the measured EC flux during the mixing period 22 –26 September 2014. Thick blue line is the median value and the light blue area shows the 25th and 75th percentiles. Gray areas represent night-time. Note the change in y-axis scale in subplot d.

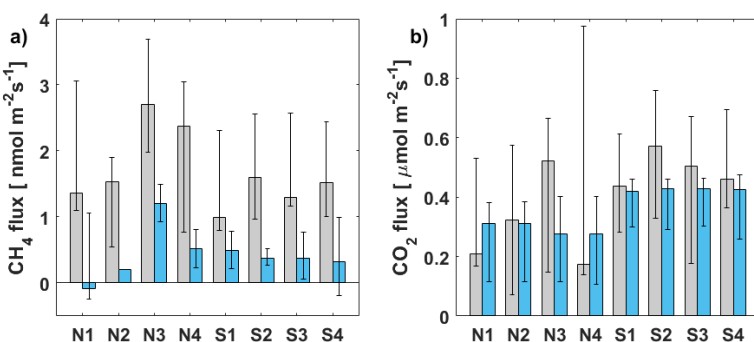

**Figure 11.** Median (a) $CH_4$ and (b) $CO_2$ FC fluxes (grey bars) at different measurement spots and median of simultaneous EC measurements (blue bars) during lake stratification. Black whiskers represent the 25th and 75th percentiles.