# Peer review of "Methane and carbon dioxide fluxes over a lake: comparison between eddy covariance, floating chambers and boundary layer method"

_Biogeosciences, 2017_

## Referee Comment (RC1) · Anonymous Referee #1 · 20 Mar 2017

Scientific significance: Good: It is not clear whether the whole data set is new, but the comparison between EC measurements and Boundary Layer Methods (BLM) offers new insights on these data

Scientific quality: Good: Applied method are valid, and the research group has a well expertise on this topic. References are appropriate, very few might be added to support assumptions (see comments)

Presentation quality: Good: English is good, figures and tables are all necessary. Some of the figures could be improved to support discussion (see comments)

+General comments:

This manuscript by Erkkilä et al deals with the important question of assessing $CO_2$ and $CH_4$ fluxes from water bodies, boreal lake in this case. This paper is fallen well in the scope of the Biogeosciences journal. MS present a set of Eddy Covariance flux measurements and BLM flux calculations, all data of interest for the research community and GHG inventory compilers. This MS is generally well written, is timely and interesting to understand the parameters of influence on carbon emissions from lakes.

The authors have done a good job in data collecting and study design though the study period is quite short (15d), but still, interesting by the contrast it is showing between stratified and mixed conditions. Some aspects of the discussion elements should be reworded to make the main conclusions more evident. Some of the conclusions, for example on the difference between day and night time, do not seem so evident based on the figure analysis. Those figures should also be improved to ease comparisons between fluxes and controlling parameters on one side, and between approaches on the other side.

+Specific comments:

-Page 2, line 7: it is Heiskanen et al (2015) rather than 2014

-Page 2, line 11: "... a small part of a lake": rather vague...

-Page 2: lines 27-29: "Because current up-scaling estimates are based on these methods, comparison is needed to reduce the uncertainties in current estimates of the role of lakes in global carbon cycle". More generally, the role of freshwaters need to better assessed.

-Page 5, line 12: with 27% and 32 data coverage for $CO_2$ and $CH_4$ fluxes are quite low, though not critically low. What is the coverage for heat fluxes? Is there any estimate of the impact of gap filling with bulk model on those fluxes?

-Page 8, lines 2-5: do you assign difference in $CO_2$ concentration between the automatic and manual systems to the same reason as for $CH_4$ (too short time of equilibra-

tion in the automatic system?). With 40min, do you really think time of equilibration was too short? What percentage of dissolved CH4 do you think you were retrieving? This issue needs to be discussed.

-Page 8, from line 23, section 3.2.1 There is a main issue here in defining detection limit, uncertainties, errors. If detection limit is approximately 2 nmol m-2 s-1, then you cannot write that "CH4 fluxes... were small (less than 1 nmol m-2 s-1)". Identically, you cannot say that "the difference between manual and automatic BLM fluxes remained below 0.4 nmol m-2 s-1". All this is not consistent. You should give indications on how you determine the flux detection limit.

-Page 8, lines 18-20: you expect an enhancement of CO2 concentration at the surface due to up-welled methane. You mean CO2 from oxidised CH4? There is at least a factor 10 between CO2 concentration at the surface and CH4 concentrations at 11m depth, so the proportion of CO2 to be expected from methane oxidation between 11m and the surface would remain low in all cases...

-Page 8, line 34: why more frequent sampling should necessarily lead to higher fluxes than the ones reported by Miettinen et al? Give explanations.

-Page 8, line 34: Give value of high fluxes reported by Ojala et al.

-Page 9, line 1: please add reference to support hypothesis on lateral CH4 transport from catchment linked with precipitation event.

-Page 9, line 3: "... that kTE and kHE were similar...": add "and comparable to EC measurements" to that sentence.

-Page 9, line 8: detail explanation in Schubert et al for lower kCC results, if relevant for this study.

-Page 9, lines 11-13: make consistent, CO2 flux or fluxes, singular/plural

-Page 9, line 16: EC increase seems be lower than a factor 3. See Table 2.

-Page 9, line 20: 3 $\mu$mol m-2 s-1: calculated from which BLM model?

-Page 9, line 25: "The same result...": that is, kCC lower than both kTE and kHE?

-Page 9, lines 28-29: again, what is the impact of using bulk formulas on the calculations of heat fluxes and subsequent kHE and kTE?

-Page 10, section 3.3. This section is rather confused. If EC is taken as the reference (line 21), then discussion on CO2 diurnal variation should try to explain why BLM show a diurnal variation which is not expected at the end, as seen from EC measurements.

-Page 10, line 6: "...kCC results in a remarkably lower flux than kHE in general": underestimation seems particularly due to underestimation of fluxes when they are at their maximum. Any reason why?

-Page 10, lines 9-10: is horizontal turbulence assumption consistent with kHE variability given in previous sentence?

-Page 10, line 15: maximum is rather reached at noon than during the afternoon

-Page 10, line 16: " The BLM flux by kTE is thus also larger in the daytime despite the lower $\Delta$[CO2]." $\Delta$[CO2] is the same for all the BLM models, why adding this element in the discussion, it is somehow confusing...

-Pages 10-11, section 3.3.2 Whole section is not convincing. Daytime vs. night time fluxes would need to be calculated to support the discussion. First define precisely hours of the day used to separate the two periods. BLM daytime fluxes do not seem to be significantly higher than night time fluxes. Diurnal variation from EC fluxes not well correlated. No bubbling? Figures not very helpful for comparison and to support discussion.

-Page 10, lines 26-27: Highest flux value is reached during the late afternoon/evening and during the second part of the night. Not so clear for EC fluxes.

-Page 10, lines 27-28: add wind speed on plot for better comparison.

-Page 10, line 29: precise which Fig 9 panel.

-Page 10, line 31-32: reword sentence: it appears that there is an increase of CH4 in the afternoon just because of less oxidation. It is both possibly due to that phenomenon and to continuous feeding of CH4 from underneath.

-Page 10, line 32: "...enhanced resuspension from sediments". do you mean lateral advection? Suspension of gases? Not clear to me. Any reference to support the assumption

-Page 10, line 33: "detached gases...": detached does not seem an appropriate word

-Page 11, line 6: any reference to support enhanced night time production of CH4 in sediments?

-Page 11, lines 9 and 11: again highest fluxes around noon seems more correct.

-Page 11, lines 13-14: Not clear, are you discussing comparison between day vs. night fluxes, or mixed/stratified periods?

-Page 11, line 16: " This may be caused by increased convective transport of CO2 from deep waters to the surface": any other reason possible?

-Page 11, section 4 There is no clear added value of this whole section to the paper. If you have floating chamber measurements and mixing ratio of CO2 and CH4 in the water, you should try to calculate a site specific k value and compare it to the one calculated from Cole and Caraco.

You use 'median' for 'standard diurnal variation' throughout the section.

-Page 11, line 28: quote reference(s) that show that anchored chambers can enhance artificially the turbulence and subsequent fluxes.

-Page 12, line 2: see comment on figure 11.

-Page 12, line 19: see comment on page 11 section 4 on calculation of site specific k

value

-Figures 2 and 3: Variation of parameters are hard to see with original figure dimensions

-Figure 3b: CH4 concentration at 11m: mention that it is the blue line.

-Figure 4: you should add in the legend, as in Figure 5: ". . . the outliers are represented with the red '+' symbol." A definition need to be given for "outliers" (>3sigma?). Some outliers seems not so different than extremes values kept in the distribution (in Figure 5 particularly), and sometimes fluxes appearing as outliers where not removed (see CH4 fluxes on September 22 and 23, panel a, or CO2 fluxes on September 14 and 15, panel b).

-Discussion on Figure 4 and 5 should be supported by statistics on difference/similarity between the different fluxes assessments.

-A different Y axis could be given for the stratified period. A dead band corresponding to the detection limit for fluxes could also be added.

-There were no measurements for CH4 from the automatic system on September 22, 23 and 24?

-Figure 5: There are no negative fluxes from BLM model, when such CO2 sink is sometime measured with the EC system. Develop to explain this major difference.

-Figure 6: add wind speed.

-Figure 6 all through fig 10: add shaded area for day/night time periods or add radiation data to better discriminate the two periods you are commenting.

-Figure 8: use same scale for fluxes calculated with kCC (up to 20 nmol m-2 s-1) It does not appear as evident that daytime fluxes are higher than night time ones. See implication for the discussion.

-Figure 11: Seems that whiskers are showing smaller flux value than what should be error on EC fluxes (see comment on page 8 about errors, precision and detection limit)

Add statistics to comment spatial variability on CH4 FC fluxes

---

## Referee Comment (RC2) · Anonymous Referee #2 · 6 Apr 2017

This manuscript compares with three widely used methods (Eddy covariance, Floating chamber and Boundary layer method) applied for measuring CH4 and CO2 flux over the boreal lake surface, which gives valuable information on measurement technique development needs. Variation of CH4 and CO2 flux during stratified and mixing period is both investigated. The results showed there is the deficiency for each method. The main question is as follows: 1) This manuscript compares CH4 and CO2 flux estimated with three BLM methods (KTE, KHE and KCC), and the linear fit results has been shown in table1 and table 2. The author considered the BLM methods with slope ratio closer to 1 as the best one method. Usually the (determination coefficient) r2 and RMSE are used for statistical criteria. As there is intercept, it's doubted that the slope

ratio could be effective. Regarding for the r2, the KTE and KHE still have higher r2, and KCC has a lower one. However the conclusion is the same.

---

## Author Comment (AC2) · 12 Jun 2017

This manuscript compares with three widely used methods (Eddy covariance, Floating chamber and Boundary layer method) applied for measuring CH4 and CO2 flux over the boreal lake surface, which gives valuable information on measurement technique development needs. Variation of CH4 and CO2 flux during stratified and mixing period is both investigated. The results showed there is the deficiency for each method. The main question is as follows: 1) This manuscript compares CH4 and CO2 flux estimated with three BLM methods (KTE, KHE and KCC), and the linear fit results has been shown in table1 and table 2. The author considered the BLM methods with slope ratio closer to 1 as the best one method. Usually the (determination coefficient) r2 and RMSE are used for statistical criteria. As there is intercept, it's doubted that the slope ratio could be effective. Regarding for the r2, the KTE and KHE still have higher r2, and KCC has a lower one. However the conclusion is the same.

We thank the anonymous referee #2 for your comment. We will add discussion about the $r^2$ value to the text.

---

## Author Response (AR1)

**Associate Editor Decision: Reconsider after major revisions** (19 Jun 2017) by Gwenaël Abril
Comments to the Author:
Decision on MS BG 2017-56

Dear authors,

I have now read your answers to the reviewer's comments and projected changes to the manuscript. Because I found reviewer's comments were at some places superficial, I made my own detailed reading of your submitted paper. Although it was clear to me that the presented dataset is of high quality for publication, my major concern comes from the lack of well-defined objectives of the paper and the justification of the choices made in the way data are analysed and presented. What are the important new insights provided by the study? How can this paper help future CO2 and CH4 fluxes measurements in lakes? These questions must be answered more clearly in your revised MS. It was not evident to me if your paper was focussed on the comparison between EC, FC and k methods, on the validity of different parameterization of k, on the diurnal variations of fluxes at the study site, or on the difference between stratified and mixed periods at the study site, etc…

Thank you for your comments. The research objectives are now made clearer and the paper focuses in comparison of the three flux measurement methods and making recommendations for future flux measurements over lakes. The new insights come from frequent sampling that allows us to examine the differences between day and night-time fluxes, with all three methods. We find that the old, widely used, gas transfer coefficient model gives substantially lower fluxes than the other methods, and there are differences between manual and automatic gas concentration measurements. All these affect the resulting flux calculation, and especially current global up-scaled flux estimates have deficiencies, as they are mostly based on the old gas transfer model and daytime concentration measurements.

To that respect, I recommend you consider very seriously the general comment by Ref.#1: "some aspects of the discussion elements should be reworded to make the main conclusions more evident. Some of the conclusions, for example on the difference between day and night time, do not seem so evident based on the figure analysis. Those figures should also be improved to ease comparisons between fluxes and controlling parameters on one side, and between approaches on the other side."

Figures are improved to make comparison between daytime and night-time more clear and in addition a new table is added where average daytime and night-time fluxes are listed for the stratified and mixing periods separately.

I also fully agree with review#1's comment "If you have floating chamber measurements and mixing ratio of CO2 and CH4 in the water, you should try to calculate a site specific k value and compare it to the one calculated from Cole and Caraco"
I would add that k could be calculated from EC and FC fluxes and compared to those calculated with the different BLM. If the paper deals with performance of methods (maybe not), calculations of k can be done separately for CO2 and CH4, normalised with the Schmidt Number and compared one with the other. This would allow a comparison of performance of methods, even when fluxes are extremely low for CH4 but not for CO2 (due to larger delta CO2 between air and water). In your respond to this comment, you mention a second paper in preparation. Your dataset is indeed strong enough to produced two papers, but splitting the data in two papers must be done according to well-defined specific objectives.

We thought about this aspect thoroughly, but came to the conclusion that we will still write two separate manuscripts. The current one is focused on flux measurement method comparison and gas fluxes in general, whereas the second one will focus more on the physics of gas transport, for which site specific k calculation is more appropriate and calculated with other methods as well.

Five Figures (6-10) report "median" diurnal variations of fluxes and driving parameters. If I understood properly, these are composite data averaging values at the same hour but different days, so these data include natural day-to-day variations. I was not convinced that using such composite median values was the best way of comparing the efficiency of the different methods, which could be done using individual synchronous values.

Daily median values of each flux are compared in Table 1. Using daily median values for comparison instead of half-hour synchronous values produces a less noisy comparison. Diurnal variation is studied for the two time periods (lake mixing and stratification) separately, because the variation considerably differs between these two periods. Figures 4 and 5 show that especially during lake stratification there is not much day-to-day variation.

To summarise, I am happy to encourage you to proceed with the revision of your manuscript, but I would like to stress that this should be done after re-think profoundly the objective of your study and the way the data are analysed. Needs some work. The paper must be re-focused. Separating the results section from the discussion section will probably help your writing.
One detail: P4L27-Final gas concentrations in the water were calculated using the Henry's Law
Please mention what solubility coefficients you used for CO2 and CH4

Solubility constants are added to the text. The manuscript is now more focused and figures are improved. We still kept results and discussion in the same section, as it helps analyzing the results more deeply. However, we have added more text on both results and discussion in the manuscript!

Looking forward to reading a revised version of you MS.

With best regards
Gwenaël Abril, BG Associate Editor
Scientific significance: Good: It is not clear whether the whole data set is new, but the comparison between EC measurements and Boundary Layer Methods (BLM) offers new insights on these data
Scientific quality: Good: Applied method are valid, and the research group has a well expertise on this topic. References are appropriate, very few might be added to support assumptions (see comments)
Presentation quality: Good: English is good, figures and tables are all necessary. Some of the figures could be improved to support discussion (see comments)

We thank the anonymous referee #1 for your very good and on point comments. We have carefully read through all the comments and prepared responses to all specific comments below.

**+General comments:**
This manuscript by Erkkilä et al deals with the important question of assessing CO2 and CH4 fluxes from water bodies, boreal lake in this case. This paper is fallen well in the scope of the Biogeosciences journal. MS present a set of Eddy Covariance flux measurements and BLM flux calculations, all data of interest for the research community and GHG inventory compilers. This MS is generally well written, is timely and interesting to understand the parameters of influence on carbon emissions from lakes. The authors have done a good job in data collecting and study design though the study period is quite short (15d), but still, interesting by the contrast it is showing between stratified and mixed conditions. Some aspects of the discussion elements should be reworded to make the main conclusions more evident. Some of the conclusions, for example on the difference between day and night time, do not seem so evident based on the figure analysis. Those figures should also be improved to ease comparisons between fluxes and controlling parameters on one side, and between approaches on the other side.

Thank you! The study period is short due to intensive manual data collection that was timed to the start of the lake mixing period. All the comments will be taken into account and the figures improved.

**+Specific comments:**
-Page 2, line 7: it is Heiskanen et al (2015) rather than 2014
Yes of course, this will be fixed.

-Page 2, line 11: ". . . a small part of a lake": rather vague. . .
Will be changed to "…is representative of the measurement point only"

-Page 2: lines 27-29: "Because current up-scaling estimates are based on these methods, comparison is needed to reduce the uncertainties in current estimates of the role of lakes in global carbon cycle". More generally, the role of freshwaters need to better assessed.
"lakes" will be changed to "freshwaters"

-Page 5, line 12: with 27% and 32 data coverage for CO2 and CH4 fluxes are quite low, though not critically low. What is the coverage for heat fluxes? Is there any estimate of the impact of gap filling with bulk model on those fluxes?

27% and 32% coverage for lake eddy covariance measurements is not that low, keeping in mind that we also have to discard data according to wind direction on top of other quality filtering. Coverage of heat fluxes was 83% and 80% for latent and sensible heat fluxes, respectively. The effect of gap-filling is probably not large, because the few gaps in heat fluxes most probably coincide with gaps in gas fluxes and will not affect the comparisons. The bulk-transfer relationships are widely used, and well-established in-situ coefficients are reported in Mammarella et al. (2015).

-Page 8, lines 2-5: do you assign difference in CO2 concentration between the automatic and manual systems to the same reason as for CH4 (too short time of equilibration in the automatic system?). With 40min, do you really think time of equilibration was too short? What percentage of dissolved CH4 do you think you were retrieving? This issue needs to be discussed.
We don't know if the equilibration time was too short or not but this a possibility. This is why we included the manual headspace samples to get a feeling for the uncertainty of the absolute values. Headspace samples were taken at different locations than the automatic system, so some variation between the two samples is expected. Without additional headspace samples close to the automatic system it is impossible to say anything absolute about the accuracy, but we still trust the gradient.

-Page 8, from line 23, section 3.2.1 There is a main issue here in defining detection limit, uncertainties, errors. If detection limit is approximately 2 nmol m-2 s-1, then you cannot write that "CH4 fluxes. . . were small (less than 1 nmol m-2 s-1)". Identically, you cannot say that "the difference between manual and automatic BLM fluxes remained below 0.4 nmol m-2 s-1". All this is not consistent. You should give indications on how you determine the flux detection limit.
Detection limit was not calculated for this site but a reference was taken from Peltola et al. (2014) to have a rough estimate of detectable flux by this analyzer. A site specific detection limit may be calculated as the standard deviation of the cross-covariance function values far from the maximum values (flux), as was developed in Wienhold et al. (1995), and added to the text. As the EC fluxes are presented as daily median values, the detection limit should be scaled by $\sqrt{n}$ where n is the number of measurements per day (n=48), giving a daily flux detection limit of approximately 0.29 nmol m-2 s-1. We will take a closer look on the detection limit and make the text more clear on this.

-Page 8, lines 18-20: you expect an enhancement of CO2 concentration at the surface due to up-welled methane. You mean CO2 from oxidised CH4? There is at least a factor 10 between CO2 concentration at the surface and CH4 concentrations at 11m depth, so the proportion of CO2 to be expected from methane oxidation between 11m and the surface would remain low in all cases...
I mean the upwelling water that would also bring CO2 to the surface from deeper waters. However, the CO2 concentration difference between surface and 11m is not as drastic as for CH4 (surface water CH4 concentration is only 3% of the 11m concentration whereas surface water CO2 concentration is 20% of the 11 m concentration) and thus upwelling water does not make such a big rise in the surface water CO2 concentration. This will be made more clear in the text.

-Page 8, line 34: why more frequent sampling should necessarily lead to higher fluxes than the ones reported by Miettinen et al? Give explanations.
More frequent sampling would include also nighttime cooling periods and perhaps windier daytime episodes in the measurements. When all kinds of weather conditions and cooling periods are included, the fluxes might end up being higher.

-Page 8, line 34: Give value of high fluxes reported by Ojala et al.

The high fluxes after rain events were about 6 nmol m-2 s-1, which is much higher than we measure during the stratified period. The value will be given in the text as well.

-Page 9, line 1: please add reference to support hypothesis on lateral CH4 transport from catchment linked with precipitation event.
Ojala et al. (2011), Rantakari & Kortelainen (2005), will be added to the text.

-Page 9, line 3: ". . . that kTE and kHE were similar. . .": add "and comparable to EC measurements" to that sentence.
Will be added.

-Page 9, line 8: detail explanation in Schubert et al for lower kCC results, if relevant for this study.
Eddy covariance and floating chambers gave 8 and 7 times higher cumulative fluxes than kCC method in Schubert et al. (2012). This will be added to the text.

-Page 9, lines 11-13: make consistent, CO2 flux or fluxes, singular/plural
Will be made consistent.

-Page 9, line 16: EC increase seems be lower than a factor 3. See Table 2.
This value is not seen from the table. Table 1 & 2 give linear fit values to the comparison between EC and BLM fluxes for CH4 and CO2 fluxes, respectively, during the whole measurement period. CO2 flux increases from about 0.5 $\mu$mol m$^{-2}$ s$^{-1}$ to 1.7 $\mu$mol m$^{-2}$ s$^{-1}$, thus a factor 3 increase.

-Page 9, line 20: 3 $\mu$mol m-2 s-1: calculated from which BLM model?
Good point, 3 $\mu$mol m$^{-2}$ s$^{-1}$ from kTE and kHE models, whereas kCC increased to 2 $\mu$mol m$^{-2}$ s$^{-1}$. This will be corrected in the text.

-Page 9, line 25: "The same result. . .": that is, kCC lower than both kTE and kHE?
"The same result…" refers to the previous sentence that compares BLM and EC methods, not the different k models.

-Page 9, lines 28-29: again, what is the impact of using bulk formulas on the calculations of heat fluxes and subsequent kHE and kTE?
Probably not large and the heat flux coverages were quite high in any case and the bulk formulas are well established.

-Page 10, section 3.3. This section is rather confused. If EC is taken as the reference (line 21), then discussion on CO2 diurnal variation should try to explain why BLM show a diurnal variation which is not expected at the end, as seen from EC measurements.
EC is not taken as a reference, but it is compared to other methods. We do not assume any method being "more correct" than others and different methods may have different diurnal variations.

-Page 10, line 6: ". . .kCC results in a remarkably lower flux than kHE in general": underestimation seems particularly due to underestimation of fluxes when they are at their maximum. Any reason why?
Early morning times are usually a time of lake cooling, which then causes some additional convective mixing in the water column. This is taken into account in kHE model, but not in kCC model.

-Page 10, lines 9-10: is horizontal turbulence assumption consistent with kHE variability given in previous sentence?

Unfortunately it is not completely clear what is meant by 'horizontal turbulence assumption' within this context. Both kCC and kHE are 1-D models considering only the vertical transport. Neither one of them is capable of addressing horizontal processes, and therefore we think that horizontal turbulence is not the cause of differences between the dynamics of these models.

-Page 10, line 15: maximum is rather reached at noon than during the afternoon

Correct, will be corrected in the text.

-Page 10, line 16: " The BLM flux by kTE is thus also larger in the daytime despite the lower _[CO2]." _[CO2] is the same for all the BLM models, why adding this element in the discussion, it is somehow confusing. . .

The higher kTE is more dominant than the lower concentration, resulting in a higher BLM flux.

-Pages 10-11, section 3.3.2 Whole section is not convincing. Daytime vs. night time fluxes would need to be calculated to support the discussion. First define precisely hours of the day used to separate the two periods. BLM daytime fluxes do not seem to be significantly higher than night time fluxes. Diurnal variation from EC fluxes not well correlated. No bubbling? Figures not very helpful for comparison and to support discussion.

Precise hours will be given in the text and shown in the figures. I would not expect much bubbling from the study area, since the lake is quite deep in the EC footprint area and from the chamber measurements we only saw one or two ebullition events close to the shore (these were removed from the analysis).

-Page 10, lines 26-27: Highest flux value is reached during the late afternoon/evening and during the second part of the night. Not so clear for EC fluxes.

True, also EC fluxes show the late afternoon peak but not the early morning peak. We will rephrase the sentence.

-Page 10, lines 27-28: add wind speed on plot for better comparison.

Wind speed is given in figures 2c, 7b and 9b.

-Page 10, line 29: precise which Fig 9 panel.

Fig 9d, will be specified in the text.

-Page 10, line 31-32: reword sentence: it appears that there is an increase of CH4 in the afternoon just because of less oxidation. It is both possibly due to that phenomenon and to continuous feeding of CH4 from underneath.

This has already been stated in line 29, but can be clarified in lines 31-32 as well.

-Page 10, line 32: ". . .enhanced resuspension from sediments". do you mean lateral advection? Suspension of gases? Not clear to me. Any reference to support the assumption

Resuspension in the littoral zone, bringing methane to lake water and then transported further in the lake by lateral advection. This is of course speculation since we do not measure it. Reference given in the text is study by Bussmann, 2005.

-Page 10, line 33: "detached gases. . .": detached does not seem an appropriate word

Will be replaced by "… enhanced transport"

-Page 11, line 6: any reference to support enhanced night time production of CH4 in sediments?

This is speculation by Podgrajsek et al. (2014), we do not observe higher nighttime CH4 fluxes. This part of the sentence can also be removed, since there is no other reference and it is not related to our study.

-Page 11, lines 9 and 11: again highest fluxes around noon seems more correct.
Yes, this will be corrected.

-Page 11, lines 13-14: Not clear, are you discussing comparison between day vs. night fluxes, or mixed/stratified periods?
Comparison between day vs. night during the mixing period. We will try to make it clearer.

-Page 11, line 16: " This may be caused by increased convective transport of CO2 from deep waters to the surface": any other reason possible?
Algal photynthesis may also play a role here although we do not have any direct result on that.

-Page 11, section 4 There is no clear added value of this whole section to the paper. If you have floating chamber measurements and mixing ratio of CO2 and CH4 in the water, you should try to calculate a site specific k value and compare it to the one calculated from Cole and Caraco.
We disagree with this and would like to keep the section. Even though we did not find any "hot spots" with the FC measurements, we think it is important to show that. With this analysis we found that EC measurements give higher CO2 fluxes from the south, shallower side of the lake and the CO2 EC fluxes did not differ from FC fluxes from the north side of the lake. Site specific k calculation was left out of the analysis for keeping the paper more compact, but we think the reviewer have a good point with suggestion this approach as well. However, there is another manuscript in preparation that is focused on gas transport and includes site specific k calculation from this campaign and we think that adding this calculation to this manuscript would make it more broad and complicated, as we try to focus on flux measurement methods instead of gas transport velocity.

You use 'median' for 'standard diurnal variation' throughout the section.
This will be corrected.

-Page 11, line 28: quote reference(s) that show that anchored chambers can enhance artificially the turbulence and subsequent fluxes.
Lorke et al. (2015) found that anchored chambers create artificial turbulence in *running* waters. This has not been found to be an issue for lakes, as studied in Gålfalk et al. (2013).

-Page 12, line 2: see comment on figure 11.
Yes.

-Page 12, line 19: see comment on page 11 section 4 on calculation of site specific k value
Yes.

-Figures 2 and 3: Variation of parameters are hard to see with original figure dimensions
Will be improved.

-Figure 3b: CH4 concentration at 11m: mention that it is the blue line.
Yes, good point.

-Figure 4: you should add in the legend, as in Figure 5: ". . . the outliers are represented with the red '+' symbol." A definition need to be given for "outliers" (>3sigma?). Some outliers seems not so different than extremes values kept in the distribution (in Figure 5 particularly), and sometimes fluxes appearing as outliers where not removed (see
CH4 fluxes on September 22 and 23, panel a, or CO2 fluxes on September 14 and 15, panel b).
Box plots come from Matlab's box plot function. It determines the extreme values (w, that are not outliers) as approximately +/- 2.7 σ and 99.3 % coverage and outliers as larger than q3+w(q3-q1) or smaller than q1-w(q3-q1) where q1 and q3 are the 25th and 75th percentiles, respectively.

-Discussion on Figure 4 and 5 should be supported by statistics on difference/similarity between the different fluxes assessments.
Statistics are already reported in Tables 1 and 2.

-A different Y axis could be given for the stratified period. A dead band corresponding to the detection limit for fluxes could also be added.
While a separate y axis might enhance readability for the stratified period, we think it would make it harder to read the whole figure. A dead band for the detection limit will be added.

-There were no measurements for CH4 from the automatic system on September 22, 23 and 24?
There were no manual measurements 22-24 Sep (Figs. 4b and 5b), but the automatic system was running.

-Figure 5: There are no negative fluxes from BLM model, when such CO2 sink is sometime measured with the EC system. Develop to explain this major difference.
Median CO2 sink was measured with the eddy covariance system only during two days but statistically these measurements were not different from 0 (tested with Mann-Whitney U-test). We can conclude that the fluxes are very small before the lake mixing. It is known that boreal lakes may occasionally act as sinks of CO2 (e.g. Huotari et al. 2011), but at this time of year it is very unlikely.

-Figure 6: add wind speed.
Wind speed is already given in Fig 7b.

-Figure 6 all through fig 10: add shaded area for day/night time periods or add radiation data to better discriminate the two periods you are commenting.
This is a good idea, a shaded area will be added.

-Figure 8: use same scale for fluxes calculated with kCC (up to 20 nmol m-2 s-1) It does not appear as evident that daytime fluxes are higher than night time ones. See implication for the discussion.
Same scale can be used, although then the diurnal variation is not that evident.

-Figure 11: Seems that whiskers are showing smaller flux value than what should be error on EC fluxes (see comment on page 8 about errors, precision and detection limit). Add statistics to comment spatial variability on CH4 FC fluxes
Coefficient of variation or standard deviation will be added.

- Former Tables 1 & 2 are merged into one Table 1
- Another Table 2 is added where the daytime and night-time average fluxes and their standard errors are listed for each measurement method separately
- The text has been modified according to reviewer and editor comments
- More description and analysis is added to section *3 Results and Discussion*, especially under subsections *3.3 Diurnal variation of estimated fluxes* and *3.4 
[revised manuscript text omitted]

---

## Author Response (AR2)

Dear authors

Because I wanted to move forward the edition of your paper, I didn't send it back to reviewers and I made my own detailed reading of your revised MS. My conclusion is that the presented dataset is of excellent quality and your conclusions are highly relevant for the community. Your paper must be published soon in BG. However, this large amount of data must be better organized, presented and synthesized in order to be published as a paper. I still find your revised MS confusing and difficult to read and unfortunately, I cannot recommend publication in its present form. When compared to the initial submission, little changes were made in the organization of the MS and some of my previous comments are still valid with this revised version. The result and discussion section is the most critical part of your MS, being often confusing and difficult to follow; fortunately the abstract and the conclusion are excellent and, together with figures Figure 2-5, they convince the reader of the quality of the presented work. However, throughout your "result and discussion section" you jump from CH4 fluxes to CO2 fluxes, from one method to another, from the stratified to mixed period, from discussion of methods to discussion of processes, etc… As you show and discuss your data at the same time, the reader often gets lost. I respect your choice to maintain results and discussion together; however, the organization of the text does not always appear logical (see below). In addition some figures are also confusing, particularly the five figures 6-10 showing diurnal variations of different averaged parameters, in many cases using different axes which complicates even more the reading and the comparison of one panel with another; The information contained in these 5 repetitive figures must be summarized in one or two figures showing the most important findings and a couple of tables with detailed statistical analysis (are fluxes for stratified/mixed, daytime/nightime and different methods, significantly different or not ?). Alternatively, these 5 figures could appear as Suppl. Material. Any change that will make the result and discussion shorter, more focussed and easier to read will be welcome.

Content of the result and discussion section is as follows:

3.1 Environmental conditions
3.1.1 Water column temperature and gas profiles
Here CH4 and CO2 are mixed
Note that there is no sub-section 3.1.2

3.2 Comparison between boundary layer model and eddy covariance flux estimates
3.2.1 CH4 fluxes
3.2.2 CO2 fluxes
3.3 Diurnal variation of estimated fluxes
3.3.1 Stratified period
3.3.2 Mixing period
3.4 Comparison between floating chambers and eddy covariance fluxes
3.4.1 CH4 fluxes
3.4.2 CO2 fluxes

We would like to express our thanks to the editor for these insightful comments and the time and dedication, which have helped to make this manuscript more focused and easier to follow. Listing the contents of the results and discussion section made us see how this manuscript may be difficult to read. We have now reorganized the text so that each gas/ gas flux is discussed in their own subsections, including all methods and diurnal and spatial variation. The contents of the Results and Discussion section are now organized as follows:

3.1 Environmental conditions and water column temperature
3.2 Water column gas concentration profiles
       3.2.1 $CH_4$ concentration profile
       3.2.2 $CO_2$ concentration profile
3.3 $CH_4$ flux comparison
       3.3.1 Spatial variation of $CH_4$
3.4 $CO_2$ flux comparison
       3.4.1 Spatial variation of $CO_2$

In addition, Figures 6, 8 and 10 were removed and replaced by Tables 1 and 3 including statistics, while Figures 7 & 9 were moved to the Appendix. We also removed subplots 4b and 5b, as they made discussion more complicated and did not affect the conclusions. We were not able to make the results and discussion much shorter, but it is now more focused easier to follow.

What is the motivation for comparing BLM and EC in 3.2 and FC and EC in 3.4, with diurnal variation in between? Why not comparing BLM with FC?
We thank the editor for pointing this deficiency out. Figures 4 & 5 were replaced by figures that show the daily median flux values measured with each method, including FC measurements. Diurnal variation is discussed together with flux comparisons. FC method is also compared in Table 2 with EC measurements.

When diurnal variation is not significant it is probably not necessary to show the curves.
All figures showing diurnal variation of fluxes were removed and replaced by Tables 1 and 3. Diurnal variation of gas transfer coefficients are shown in Appendix figures.

P8 section 3.11 and throughout the whole MS: Concentration units are sometimes mmol m-3 and sometimes nmol m-3. I find strange µmol m-3 never appear. Please check. I suggest choosing one single concentration unit for each gas.
We thank the editor for noticing this error. All concentration units were changed to mmol m$^{-3}$ for clarity.

P8
L20 : "equilibration time of 40 min should be enough" : any objective evidence for that?
We measured $CO_2$ concentration in the lake surface with two different automatic systems that agreed well each other as well as manual samples after 16 Sept. Since three different methods compare quite well with each other and $CO_2$ is more soluble in water than $CH_4$, we can assume that 40 min equilibration time is enough for $CO_2$. The whole sentence now reads "$CO_2$ is more soluble in water than $CH_4$ and thus equilibration time of 40 min should be enough for automatic $CO_2$ measurements *and two different automatic systems compared well with each other on $CO_2$ concentration at the surface (results not shown).* We thereby conclude the difference between automatic and manual $CO_2$ concentration measurements to be caused by spatial variation rather than the measurement system."

L22-24: discuss the effect of concentration on calculated BLM fluxes

Discussion on the effect of concentration difference was added and the sentence now reads "We point out, however, that choosing the measurement method as well as the measurement spot has an effect on the observed concentrations *and thus fluxes calculated with the BLM method as larger concentration difference between the water surface and air would result in a larger flux in general*"

L26-27 "rapid increase in the surface water concentration according to manual sample" rephrase

The sentence was rephrased and now reads: "On 14 September, surface layer mixing reached 7 m depth and brought $CO_2$ rich water from deeper waters to the surface causing a drop in $CO_2$ concentration at 7 m depth and *manual samples show a rapid increase in the surface water concentration*."

L30 "agree better" : how much ?

The sentence was reworded and now reads: "After 16 September, the automatic and manual $CO_2$ concentration measurements agree better with each other, *as the average difference between the measured concentrations decreases from 114 to 16 mmol $m^{-3}$*."

P9

L14 "higher than" refer to a table with statistical analysis that compare all types of fluxes

Statistical analysis comparing all methods to EC was added to Table 2. Manual BLM fluxes were removed, as it complicated the discussion and did not change the results.

P10

L14-15 "tested with mann-whitney U test" refer to a table with statistical analysis that compare all types of fluxes

Statistical analysis was added to Table 2.

L18 (about 60%) be precise

Sentence now reads "Linear fit parameters for the comparison of BLM *and FC methods* with EC measurements show that $k_{TE}$ ($r^2$=0.26) and $k_{HE}$ ($r^2$=0.27) give the best results when compared with EC *(60% of the measured EC flux)*"

L25 CH4 appears in a section dedicated to CO2

This section was moved to Conclusions, since it gives recommendations for future measurements and is not related to $CO_2$ fluxes exclusively.

L28-35: These important discussion statements appear very diluted within the more systematic description of results appearing above.

This section was moved to Conclusions, since it gives some recommendations for future measurements and is relevant for both $CH_4$ and $CO_2$ flux measurements.

P11

L4-6. Is comparison between methods the only interest for describing diurnal variation?

Due to reorganization of the text, this sentence is now removed and diurnal variation is discussed together with general flux level and method comparison discussion.

L14 "negligible" : provide a value
The convective term in $k_{HE}$ is zero during daytime. The sentence now reads: "Also the convective term ($C_2 w_*$) in $k_{HE}$ is *zero* during daytime when the lake is heating due to higher air temperature, resulting in a lower $k_{HE}$ (Fig. A1a)."

L16 "the convective term in kHE increases toward night-time causing higher total kHE" confusing, rephrase
The sentence was rephrased and now reads: "This is seen in Fig. A1a as the convective term $C_2 w_*$ increases towards night-time causing higher *gas transfer coefficient $k_{HE}$ and thus higher flux as well.*"

L22-23: "highest fluxes at noon when also friction velocity gains its maximum value" also when CO2 concentration was lower? The convective term is on a different scale in fig7, how much does it contribute at max and on average ?
Yes, the dominant effect seems to be the gas transfer coefficient and not concentration difference in this case. The contribution of the different terms to total gas transfer coefficient cannot be directly calculated, as the both (shear and convective) terms are summed under a 4th root (Eq. 8). However, as convection has a minimum value of 0, the shear term must have a maximum contribution of 100 % and minimum contribution from convective term would be 0 %.

L27 "lower u*w" lower than what ?
Sentence now reads "Friction velocity calculated from wind speed measurements (with a drag coefficient 0.001 for a water surface) instead of direct $u_{*a}$ measurements gave similar diurnal variation as model $k_{HE}$ (data not shown), but resulted in a lower $u_{*w}$ *than with direct $u_{*a}$ measurements*."

P12
"because the afternoon flux peak is also seen in the BLM by kCC, we can deduce that it is due to higher wind speed and enhanced shear during the afternoon as well as CH4 concentration difference": you have CH4 concentration and wind speed data, no need to deduce from the value of kCC
Good point, sentence now reads "*Higher daytime fluxes are expected due to higher wind speed and enhanced shear during the afternoon (Bastviken et al., 2010)* as well as higher $\Delta[CH_4]$, that is also partly due to enhanced mixing bringing $CH_4$ from deeper waters"

L7-8: because you are discussing methods and processes at the same time, the text becomes hard to follow. Also cite Dumestre et al. 1999 Influence of Light Intensity on Methanotrophic Bacterial Activity in Petit Saut Reservoir, French Guiana APPLIED AND ENVIRONMENTAL MICROBIOLOGY,0099-2240/99/$04.0010Feb. 1999, p. 534–539
"the larger cc difference toward the afternoon may be caused by higher oxidation rate in the dark … during night" rephrase
Citation was added and the sentence now reads "*We find lower concentration difference $\Delta[CH_4]$ in night-time that may be caused by* higher oxidation rate in dark that lowers $CH_4$ concentration in the water (Mitchell et al., 2005; *Dumestre et al., 1999*). During daytime solar radiation, the oxidation rate would then be lower resulting in an increase of water $CH_4$ concentration towards the afternoon."

L14-15 "all models give similar diurnal patterns… only magnitude are different". Ok, but this is due to the predominant effect of changes in CH4 concentrations

The editor brings up a good point, sentence was removed.

L17-21: discussion in the light of literature could be strengthened if it did not appear only as separated statements at the end of each paragraph.

Discussion was added within the text as well, not just at the end of paragraphs.

L29 "which is then visible" rephrase being more precise

Sentence now reads "Shear terms $C_1U$ and $u_*^3/(kz)$ in $k_{HE}$ and $k_{TE}$ models, respectively, have diurnal variations with highest values at noon as well (Figs. A2a and A2c), *which results in higher daytime BLM fluxes with $k_{HE}$ and $k_{TE}$.*"

L30-31, indeed, but kCC agrees with EC during part of the day. How relevant are these comparisons based on averages of measurement during 5 following days (figures 6,8 & 10): what about variation from one day to another?

The editor makes a good argument, indeed BLM $k_{CC}$ fluxes are sometimes closer to EC than other BLM models. The sentence now reads: "BLM by $k_{CC}$, however, shows considerably lower fluxes than $k_{HE}$ and $k_{TE}$ both during daytime and night-time on average. *Average daytime and night-time BLM $k_{CC}$ fluxes are closer to EC measurements than other BLM models, but do not agree well with EC on daily scale (Fig. 5).*"

P13
L 3-5 "using selectively only daytime gas concentration… global budget makes a biased assumption" AND "the EC… no clear diurnal variation during this period either". This looks like a contradiction

Sentence about EC was removed, as it has been already stated that EC measurements do not detect diurnal variation. On average these methods go well together, because the fluxes are measured both night and day. Using only daytime measurements would make daily median BLM fluxes higher and thus not comparable to EC measurements. Then again, we cannot know for sure which method is more correct, but measurements done at different times in a day will get us closer to the truth.

L14 "and the coefficient of variation was…" provide average +/- coefficient of variation and avoid such phrasing
This discussion was removed and the information is given in tables 1 and 3.

L19 "partly this difference is of course due to FC fluxes averaged over the different measurements spots… " rephrase
This sentence was removed to avoid confusion.

L24 "low fluxes are difficult to detect… close to the detection limit of the gas analyser used in the EC measurements" The detection limits of a gas analyser concerns concentration and not fluxes. Gas analysers are able to measure standard atmospheric concentration. Here you may reach the limit for EC fluxes calculation, depending on various classical criteria of EC data processing (spectral analysis, stability, etc...) Please provide this information.

We thank the editor for noticing this. Definition of the detection limit is already discussed in the Methods section, but this erroneous statement got lost in the text. The sentence now reads: "A reason behind the result might be that these low fluxes are very difficult to detect with the EC method, since the $CH_4$ fluxes were very close to *the detection limit of the EC measurement system."*

L25 "could have probably produced a better comparison" unclear statement
The sentence now reads: "*Higher fluxes during the mixing period could have been more suitable for a comparison between the two methods."*

L28 "statistically different" provide a table with complete statistical analysis. What means "to detect reliability"?
Statistical difference is discussed before in the text and results of U-test are provided in Table 2. "In this study EC and FC $CH_4$ fluxes did not compare well with each other and the difference in fluxes is statistically significant, mainly due to low $CH_4$ fluxes for the EC method to detect reliably *(well above the detection limit of the system)"*

L19 "larger source area" rephrase
The sentence now reads: "EC method has a larger source area *(flux footprint)* than FC method, which might also affect the flux."

P14
L4 "differed from daytime EC fluxes" provide a table with detailed statistical analysis
Table 3 provided and discussion moved earlier in the text.

L8-10 referring to literature elsewhere only at the end of paragraphs makes the discussion superficial
References added to middle of paragraphs as well.

L12: provide a table with detailed statistical analysis
This analysis is not listed on a table, as there would be only few parameters stating the differences/similarities of EC fluxes measured from the south/north side of the lake. Statistics are listed in the text and different methods are compared in Table 2.

L17 you are discussing CH4 fluxes in a section dealing with CO2 fluxes
This is intentional, as one might wonder why we detect only high $CO_2$ fluxes from south and not $CH_4$.

L21 "this is due to limitations in the EC method" do you mean "this is one of the limitations…"?
Yes, good point, corrected in the text as "This is due *to one of the limitations* in the EC method, because it requires a homogeneous surface and favourable wind conditions, but leads to possibly biased flux estimations, especially if flux is only measured over a particularly deep or shallow area *not representative of the lake*."

To summarize, I find your work and conclusions of excellent scientific quality, but your MS needs substantial improvement so readers can access more easily to your dataset and conclusions. I will be happy to receive a new MS with substantial revision of the results and discussion section, figures and tables.

We thank the editor again for all these insightful and helpful comments, making our manuscript hopefully more clear and easier to read.

Best Regards
Gwenaël Abril

[revised manuscript text omitted]